# SiGeo: Sub-One-Shot NAS via Information Theory and Geometry of Loss Landscape

## Abstract

Neural Architecture Search (NAS) has become a widely used tool for automating neural network design. While one-shot NAS methods have successfully reduced computational requirements, they often require extensive training. On the other hand, zero-shot NAS utilizes training-free proxies to evaluate a candidate architecture's test performance but has two limitations: (1) inability to use the information gained as a network improves with training and (2) unreliable performance, particularly in complex domains like RecSys, due to the multi-modal data inputs and complex architecture configurations. To synthesize the benefits of both methods, we introduce a "sub-one-shot" paradigm that serves as a bridge between zero-shot and one-shot NAS. In sub-one-shot NAS, the supernet is trained using only a small subset of the training data, a phase we refer to as "warm-up." Within this framework, we present SiGeo, a proxy founded on a novel theoretical framework that connects the supernet warm-up with the efficacy of the proxy. Extensive experiments have shown that SiGeo, with the benefit of warm-up, consistently outperforms state-of-the-art NAS proxies on various established NAS benchmarks. When a supernet is warmed up, it can achieve comparable performance to weight-sharing one-shot NAS methods, but with a significant reduction ($\sim 60\%$) in computational costs.

## 1 Introduction

In recent years, Neural Architecture Search (NAS) has emerged as a pivotal paradigm for automating the design of neural networks. One-shot NAS simplifies the process of designing neural networks by using a single, comprehensive supernet (i.e. search space), which contains a versatile choice of candidate architectures. This approach allows for quick evaluation of various network architectures, saving both time and computational resources. It has been successfully applied to the vision domain (Liu et al., 2018; Real et al., 2017; Cai et al., 2020; Chitty-Venkata et al., 2022; Fedorov et al., 2022) and RecSys domain (Krishna et al., 2021; Zhang et al., 2023; Song et al., 2020). One-shot NAS speeds up the process of finding the best neural network design, but it still requires training (Li et al., 2023). To mitigate this, zero-shot Neural Architecture Search (zero-shot NAS) has been developed with the aim of bypassing the need for extensive training and evaluation. The core of zero-shot NAS is to use a zero-cost metric to quickly evaluate the actual performance of candidate architectures, While a wide range of zero-shot proxies have been introduced, a recent study (White et al., 2022) has suggested that many of them do not fully utilize the information gained as a network improves with training. As a result, the pretraining of a supernet does not necessarily translate into enhanced search performance, leaving an unexplored opportunity for improvement.

Despite significant advances in zero-shot NAS in the computer vision domain, its application to other complex domains, such as recommendation systems (RecSys), remains largely underexplored. RecSys models introduce unique challenges, primarily owing to their multi-modal data inputs and the diversity in their architectural configurations (Zhang et al., 2023). Unlike vision models, which generally rely on homogeneous 3D tensors, RecSys handle multi-modal features, a blend of 2D and 3D tensors. Additionally, vision models predominantly utilize convolutional layers, whereas RecSys models are heterogeneous within each stage of the model using a variety of building blocks, including but not limited to Sum, Gating, Dot-Product, and Multi-Head Attention. This added complexity makes the naive deployment of state-of-the-art (SOTA) zero-shot NAS approaches less effective in the recommendation context.

Table 1: Comparison of SiGeo v.s. existing NAS methods for RecSys

| Method | Setting | Criteo Log Loss | Avazu Log Loss | KDD Log Loss | GPU Days |
|---|---|---|---|---|---|
| PROFIT (Gao et al., 2021) | One-shot | 0.4427 | 0.3735 | - | ∼0.5 |
| AutoCTR (Song et al., 2020) | One-shot | 0.4413 | 0.3800 | 0.1520 | ∼0.75 |
| NASRecNet (Zhang et al., 2023) | One-shot | 0.4395 | 0.3736 | 0.1487 | ∼0.3 |
| ZiCo (Li et al., 2023) | Zero-shot | 0.4404 | 0.3770 | 0.1486 | ∼0.11 |
| SiGeo (Ours) | Zero-Shot | 0.4404 | 0.3750 | 0.1486 | ∼0.11 |
| | Sub-one-shot | 0.4396 | 0.3741 | 0.1484 | ∼0.12 |

To mitigate the limitations of existing NAS methods, this work presents a novel sub-one-shot search strategy. As a setting between zero-shot and one-shot, sub-one-shot NAS allows for limited training of supernet with a small portion of data while still employing a training-free proxy for performance prediction of sampled subnets, as illustrated in Fig. 1. Therefore, the sub-one-shot NAS considers a trade-off between computational efficiency and predictive performance. In order to effectively utilize the additional information inherited from the pretrained supernet, we further develop a new **S**ub-one-shot proxy based on the **i**nformation theory and **Geo**metry of loss landscape, named **SiGeo**. In sum, this work makes the following contributions:

- We introduce SiGeo, a novel proxy proven to be effective in NAS when the candidate architectures are warmed up.
- We theoretically analyze the geometry of loss landscapes and demonstrate the connection between the warm-up of supernet and the effectiveness of the SiGeo.
- Sub-one-shot setting is proposed to bridge the gap between zero-shot and one-shot NAS.
- Extensive experiments are conducted to assess the effectiveness of SiGeo in both CV and RecSys domains under the sub-one-shot setting. The results show that as we increase the warm-up level, SiGeo's scores align more closely with test accuracies. Compared with the weight-sharing one-shot NAS, SiGeo shows comparable performance but with a significant reduction in computational costs, as shown in Table 1.

## 2 PROBLEM DESCRIPTION AND RELATED WORK

**One-Shot NAS** One-shot NAS algorithms (Brock et al., 2018) train a weight sharing supernet, which jointly optimizes subnets in neural architecture search space. Then this supernet is used to guide the selection of candidate architectures. Let $\mathcal{A}$ denote the architecture search space and $\Theta$ as all learnable weights. Let $\ell$ denote the loss function. The goal of one-shot training is to solve the following optimization problem under the given resource constraint $r$:

$$\Theta^\star = \arg\min_{\Theta} \mathbb{E}_{\boldsymbol{a}\sim\mathcal{A}}[\ell(\boldsymbol{a}|\Theta_{\boldsymbol{a}}, \mathcal{D})]$$
$$\boldsymbol{a}^\star = \arg\max_{\boldsymbol{a}\in\mathcal{A}} \mathcal{P}_{score}(\boldsymbol{a}|\Theta_{\boldsymbol{a}}^\star; \mathcal{D}_{val}) \quad \text{s.t.} \quad R(\boldsymbol{a}) \leq r$$

where $\boldsymbol{a}$ is the candidate architecture (i.e. subnet) sampled from the search space $\mathcal{A}$, $\Theta_{\boldsymbol{a}} \in \Theta$ is the corresponding part of inherited weights (i.e. subnet) and $R(\cdot)$ represents the resource consumption, i.e. latency and FLOPs. $\mathcal{P}_{score}$ evaluates performance on the validation dataset. There are many methods proposed for CV tasks (Chen et al., 2019; Dong & Yang, 2019; Cai et al., 2019; Stamoulis et al., 2019; Li et al., 2020; Chu et al., 2021; Guo et al., 2020; Fedorov et al., 2022; Banbury et al., 2021), natural language processing (So et al., 2019; Wang et al., 2020b), and RecSys task (Gao et al., 2021; Krishna et al., 2021; Zhang et al., 2023; Song et al., 2020).

**Zero-Shot NAS** Zero-shot NAS proxies have been introduced, aiming to predict the quality of candidate architectures without the need for any training. Zero-shot NAS can be formulated as

$$\boldsymbol{a}^\star = \arg\max_{\boldsymbol{a}\in\mathcal{A}} \mathcal{P}_{zero}(\boldsymbol{a}|\Theta_{\boldsymbol{a}}) \quad \text{s.t.} \quad R(\boldsymbol{a}) \leq r$$

where $\mathcal{P}_{zero}$ is a zero-shot proxy function that can quickly measure the performance of any given candidate architecture without training. Some existing proxies measure the expressivity of a deep neural network (Mellor et al., 2021; Chen et al., 2021; Bhardwaj et al., 2022; Lin et al., 2021a) while many proxies reply on gradient of network parameter (Abdelfattah et al., 2021; Lee et al., 2019; Tanaka et al., 2020; Wang et al., 2020a; Lopes et al., 2021). Recently, Li et al. (2023) introduced

ZiCo, a zero-shot proxy based on the relative standard deviation of the gradient. This approach has been shown to be consistently better than other zero-shot proxies.

**Sub-One-Shot NAS** For the weight-sharing NAS, one key difference between one-shot and zero-shot settings is the warm-up phase. In a one-shot NAS, the supernet is trained using a large dataset. However, as noted by (Wang et al., 2023), "training within a huge sample space damages the performance of individual subnets and requires more computation to search". Conversely, the zero-shot setting omits this warm-up phase, leading to a more efficient process. Nonetheless, the performance of zero-shot proxies is often less reliable and subject to task-specific limitations. To bridge the gap, we propose a new setting called "sub-one-shot", where the supernet is allowed to be warmed up by a small portion of training data (e.g. 1% training samples) prior to the search process. Our goal is to propose a new proxy that can work in both zero-shot and sub-one-shot settings; see Fig. 1a.

**Other Hybrid NAS** Recently, Wang et al. (2023) proposed PreNAS to reduce the search space by a zero-cost selector before performing the weight-sharing one-shot training on the selected architectures to alleviate gradient update conflicts (Gong et al., 2021).

**Remark 1.** Compared with other zero-shot proxies, SiGeo allows for the supernet warm-up. In practice, the warm-up with 1-10% data is enough to make SiGeo achieve the SOTA performance.

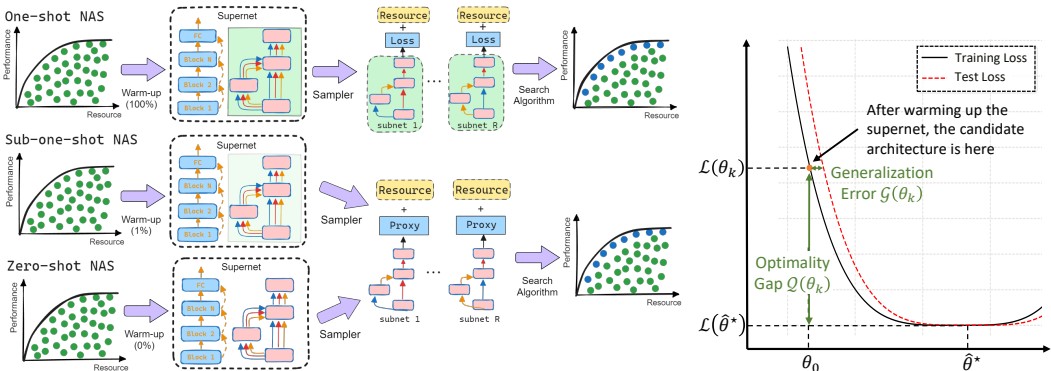

(a) Illustration of SiGeo-NAS under different NAS settings.    (b) SiGeo: Loss landscape geometry.

Figure 1: (a) SiGeo-NAS samples candidate architectures in the search space with a light warm-up. In comparison to the resource-intensive one-shot training of the supernet, this strategy offers an efficient and cost-effective means of evaluating subnets within evolution search or reinforcement learning (RL) search. In contrast to the conventional zero-shot approach, SiGeo-NAS attains notably improved performance while incurring only a minimal warm-up cost. (b) The connection between SiGeo and the geometry of loss landscape around the local minimum. The test error is approximated by the minimum achievable training loss and the generalization error.

**Information Matrix and Fisher-Rao Norm** Define $q : \mathcal{X} \times \mathcal{Y} \to \mathbb{R}$ as the probability function of *true* data distribution. The probability function of the model distribution is defined as $p_{\boldsymbol{\theta}}(\boldsymbol{x}, \boldsymbol{y}) = q(\boldsymbol{x})p_{\boldsymbol{\theta}}(\boldsymbol{y}|\boldsymbol{x})$, with the marginal input distribution $q(\boldsymbol{x})$ and the conditional distribution $p_{\boldsymbol{\theta}}(\boldsymbol{y}|\boldsymbol{x})$. Fisher information matrix $\boldsymbol{F}$ and the Hessian $\boldsymbol{H}$ are define as (Thomas et al., 2020),

$$\boldsymbol{F}(\boldsymbol{\theta}) = \mathbb{E}_{p_{\boldsymbol{\theta}}}\left[\frac{\partial^2\ell(\boldsymbol{\theta};\boldsymbol{x},\boldsymbol{y})}{\partial\boldsymbol{\theta}\partial\boldsymbol{\theta}^\top}\right] = \mathbb{E}_{p_{\boldsymbol{\theta}}}\left[\frac{\partial}{\partial\boldsymbol{\theta}}\ell(\boldsymbol{\theta};\boldsymbol{x},\boldsymbol{y})\frac{\partial}{\partial\boldsymbol{\theta}}\ell(\boldsymbol{\theta};\boldsymbol{x},\boldsymbol{y})^\top\right] \text{ and } \boldsymbol{H}(\boldsymbol{\theta}) = \mathbb{E}_q\left[\frac{\partial^2\ell(\boldsymbol{\theta};\boldsymbol{x},\boldsymbol{y})}{\partial\boldsymbol{\theta}\partial\boldsymbol{\theta}^\top}\right] \quad (1)$$

It has been shown that Fisher and Hessian matrices are identical when the learned model distribution becomes the same as the true sampling distribution (Lee et al., 2022; Karakida et al., 2019; Thomas et al., 2020). Thomas et al. (2020) has shown that these two matrices are empirically close as training progresses. The Fisher-Rao (FR) norm, defined as $\|\boldsymbol{\theta}\|_{fr} := \boldsymbol{\theta}\boldsymbol{F}(\boldsymbol{\theta})\boldsymbol{\theta}$, has been proposed as a measure of the generalization capacity of deep neural networks (DNNs) Liang et al. (2019).

**Notation.** Here we briefly summarize the notations in the paper. For any matrix $\boldsymbol{A}$, we use $\|\boldsymbol{A}\|$, and $\|\boldsymbol{A}\|_F$ to denote its operator norm and Frobenius norm, respectively. In addition, the eigenvalues $\lambda_i(\boldsymbol{A})$ of matrix $\boldsymbol{A}$ are sorted in non-ascending order, i.e. $\lambda_1 \geq \lambda_2 \geq \ldots \geq \lambda_d$. Unless otherwise specified, we use $\mathbb{E}[\cdot]$ to denote the expectation over the joint distribution of all samples accumulated in the training process to date, often referred to as the filtration in stochastic approximation literature. In what follows, we use $\oslash$ to denote the element-wise division and $\nabla$ to represent $\nabla_{\boldsymbol{\theta}}$.

## 3 CONVERGENCE AND GENERALIZATION

After the warm-up phase (i.e. the supernet is trained using a small subset of the training data), the weight of the supernet will escape the saddle points and reside near a certain local minimum denoted by $\hat{\theta}^\star$ (Fang et al., 2019; Kleinberg et al., 2018; Xu et al., 2018). This observation motivates us to focus on the convergence phase of learning algorithms, leading directly to the assumption **A.2**.

For each candidate architecture $a$ associated with the weight $\theta := \Theta_a \in \Theta$, consider a supervised learning problem of predicting outputs $y \in \mathcal{Y}$ from inputs $x \in \mathcal{X}$. Let $\hat{\theta}^\star$ denote the local minimum near which the weight of subnet is located. Given $N$ i.i.d training samples $\mathcal{D} = \{(x_n, y_n); n = 1, ..., N\}$, the training objective is to minimize the empirical training loss starting with an initial weight $\theta_0$ inherited from the pretrained supernet as follows

$$\hat{\theta}^\star \in \underset{\theta \in \mathbb{R}^d}{\arg\min}\, \ell(\theta; \mathcal{D}) := \mathbb{E}_{\hat{q}}[\ell(\theta; x, y)] = \frac{1}{N}\sum_{n=1}^{N}\ell(\theta; x_n, y_n) \tag{2}$$

where $\hat{q}$ is the empirical distribution and $\ell(\theta; x, y)$ is the loss function. Define the expected loss as $\mathcal{L}(\theta) = \mathbb{E}_q[\ell(\theta; x, y)]$, which can be decomposed at a true local minimum $\theta^\star = \arg\min_{\theta \in \mathbb{R}^d} \mathcal{L}(\theta)$:

$$\mathcal{L}(\theta^\star) = \underbrace{\mathbb{E}_q\left[\ell(\theta^\star; x, y) - \ell(\hat{\theta}^\star; x, y)\right]}_{\text{Excess Risk}} + \underbrace{\mathbb{E}_q\left[\ell(\hat{\theta}^\star; x, y)\right] - \mathbb{E}_{\hat{q}}\left[\ell(\hat{\theta}^\star; x, y)\right]}_{\text{Generalization Error}} + \underbrace{\mathbb{E}_{\hat{q}}\left[\ell(\hat{\theta}^\star; x, y)\right]}_{\text{Training Loss}} \tag{3}$$

where the first term represents the *excess risk*, the difference between test error and the minimum possible error. The second term is *generalization error* measuring the difference between train and test error and indicating the extent to which the classifier may be overfitted to a specific training set. The last term is the *training loss*. We aim to derive a proxy that measures $\mathcal{L}(\theta^\star)$ using training-free estimators of the training loss and generalization error.

To facilitate the analysis, we introduce assumptions (**A.2**-**A.4**) that the initial weight inherited from the supernet remains within a compact space, where the training loss is differentiable and has positive definite Hessian around $\hat{\theta}^\star$. We study the asymptotic behavior of iterations of the form

$$\theta_{k+1} \leftarrow \theta_k - \eta_k B_k^{-1}\nabla\ell(\theta_k; \mathcal{D}_k) \tag{4}$$

where $B_k$ is a curvature matrix and $\nabla\ell(\theta_k; \mathcal{D}_k) = \frac{1}{|\mathcal{D}_k|}\sum_{(x_n, y_n) \in \mathcal{D}_k}\nabla\ell(\theta_k; x_n, y_n)$ is a sample gradient estimate from a uniformly sampled mini-batch $\mathcal{D}_k \subseteq \mathcal{D}$ at iteration $k$. Eq. 4 becomes a first-order stochastic gradient descent (SGD) if the curvature function is an identity matrix, $B_k = I$. It turns out to be a natural gradient descent if the curvature function is the exact Fisher information matrix $B_k = F$ and second-order optimization if $B_k$ is the Hessian (Martens, 2020).

### 3.1 REGULARITY CONDITIONS

We summarize the assumptions for the regularity of stochastic gradient descent, serving as the foundation for our analysis. The justification for the assumption can be found in Appendix A.

**A.1** (Realizability) The true data-generating distribution is in the model class.

**A.2** (Initiation) The initial weight of the candidate architecture, $\theta_0$, (probably after the warm-up) is located within a compact set $\mathbb{B}$ centered at $\hat{\theta}^\star$, such that $\|\theta_0 - \hat{\theta}^\star\|_1 \leq M$.

**A.3** (Differetiability) The loss function of candidate network $\ell(\theta; x, y)$ is differetiable almost everywhere on $\theta \in \mathbb{B}$ for all $(x, y) \in \mathcal{X} \times \mathcal{Y}$.

**A.4** (Invertibility) The Hessian, Fisher's information matrix and covariance matrix of the gradient are positive definite in the parametric space $\mathbb{B}$ such that they are invertible.

### 3.2 ON CONVERGENCE OF TRAINING LOSS

Now let's analyze the convergence rate for a given candidate architecture and investigate the impact of gradient variance. Let $\mathcal{Q}(\theta) := \mathcal{L}(\theta) - \mathcal{L}(\hat{\theta}^\star) = \mathbb{E}[\ell(\theta; x, y)] - \mathbb{E}[\ell(\hat{\theta}^\star; x, y)]$ denote the (local) optimality gap from the local minimum loss. The following theorem presents the rate of convergence by using a similar technique as Garrigos & Gower (2023, Theorem 5.3.).

**Theorem 1.** *Assume A.1-A.4. Consider $\{\boldsymbol{\theta}_k\}_{k\in\mathbb{Z}}$ a sequence generated by the first-order (SGD) algorithm (4), with a decreasing sequence of stepsizes satisfying $\eta_k > 0$. Let $\sigma_k^2 := \mathrm{Var}[\nabla\ell(\boldsymbol{\theta}_k; \mathcal{D}_k)]$ denote the variance of sample gradient, where the variance is taken over the joint distribution of all the training samples until the $k$-th iteration. Then It holds that*

$$\mathbb{E}[\mathcal{L}(\bar{\boldsymbol{\theta}}_k)] - \mathcal{L}(\hat{\boldsymbol{\theta}}^\star) \leq \frac{\|\boldsymbol{\theta}_0 - \hat{\boldsymbol{\theta}}^\star\|}{2\sum_{i=0}^{k-1}\eta_i} + \frac{1}{2}\sum_{i=0}^{k-1}\sigma_k^2$$

The proof of Theorem 1 is provided in Appendix B. The key insight from this theorem is the relationship between the optimality gap and the gradient variance. In particular, the smaller the gradient variance across different training samples, the lower the training loss the model converges to; i.e., the network converges at a faster rate. In Section 3.4, we'll further discuss how the gradient variance is related to the Hessian matrix around local minima, thereby representing the curvature of the local minimum as shown in (Thomas et al., 2020).

Owing to the page limit, a comprehensive study on the convergence rate under the strong convexity assumption can be found in Appendix C. Beyond the gradient variance, this study provides insight into the relationship between the rate of convergence and the FR norm. Specifically, Theorem4 in Appendix C illustrates that a higher FR norm across training samples leads to a lower convergence point in training loss, signifying a quicker convergence rate.

### 3.3 LOWER BOUND OF MINIMUM ACHIEVABLE TRAINING LOSS

This section examines the loss landscape to provide insights into predicting the minimum achievable training loss for a candidate architecture. The discussion follows the similar analysis technique used by Martens (2020). Applying Taylor's approximation to the expected loss function $\mathcal{L}(\boldsymbol{\theta})$ gives

$$\mathcal{L}(\boldsymbol{\theta}_k) - \mathcal{L}(\hat{\boldsymbol{\theta}}^\star) = \frac{1}{2}(\boldsymbol{\theta}_k - \hat{\boldsymbol{\theta}}^\star)^\top \boldsymbol{H}(\hat{\boldsymbol{\theta}}^\star)(\boldsymbol{\theta}_k - \hat{\boldsymbol{\theta}}^\star) + \nabla\mathcal{L}(\hat{\boldsymbol{\theta}}^\star)^\top(\boldsymbol{\theta}_k - \hat{\boldsymbol{\theta}}^\star) + \mathcal{O}\left((\boldsymbol{\theta}_k - \hat{\boldsymbol{\theta}}^\star)^3\right)$$

$$= \frac{1}{2}(\boldsymbol{\theta}_k - \hat{\boldsymbol{\theta}}^\star)^\top \boldsymbol{H}(\hat{\boldsymbol{\theta}}^\star)(\boldsymbol{\theta}_k - \hat{\boldsymbol{\theta}}^\star) + \mathcal{O}\left((\boldsymbol{\theta}_k - \hat{\boldsymbol{\theta}}^\star)^3\right) \quad (5)$$

where the last inequality holds due to $\nabla\mathcal{L}(\hat{\boldsymbol{\theta}}^\star) = 0$. By taking the expectation over the joint distribution of all samples used for training until $k$-th iteration, the optimality gap (Eq. 5) becomes

$$\mathcal{Q}_k := \mathbb{E}[\mathcal{L}(\boldsymbol{\theta}_k)] - \mathcal{L}(\hat{\boldsymbol{\theta}}^\star) = \frac{1}{2}\mathbb{E}\left[(\boldsymbol{\theta}_k - \hat{\boldsymbol{\theta}}^\star)^\top \boldsymbol{H}(\hat{\boldsymbol{\theta}}^\star)(\boldsymbol{\theta}_k - \hat{\boldsymbol{\theta}}^\star)\right] + \mathbb{E}\left[\mathcal{O}\left((\boldsymbol{\theta}_k - \hat{\boldsymbol{\theta}}^\star)^3\right)\right]. \quad (6)$$

By assuming the local convexity (**A.4**) and some mild conditions, it holds $\mathbb{E}\left[\|\boldsymbol{\theta}_k - \hat{\boldsymbol{\theta}}^\star\|^2\right] = \mathcal{O}(\frac{1}{k})$; see details in Lacoste-Julien et al. (2012, Theorem 4) or Bottou & Le Cun (2005, Theorem A3). It implies that the higher order term $\mathbb{E}\left[(\boldsymbol{\theta}_k - \hat{\boldsymbol{\theta}}^\star)^3\right]$ would shrink faster, that is, $\mathbb{E}\left[(\boldsymbol{\theta}_k - \hat{\boldsymbol{\theta}}^\star)^3\right] = o(\frac{1}{k})$ (Martens, 2020). As a result, it is sufficient to focus on a quadratic loss and then present the lower bound of the minimum achievable training loss; see the proof of Theorem 2 in Appendix D.

**Theorem 2.** *Assume A.1-A.4. Let $\mu_k = \sum_{i=0}^{k}\mathbb{E}\left[\|\nabla\ell(\boldsymbol{\theta}_i; \mathcal{D}_i)\|_1\right]$ denote the sum of the expected absolute value of gradient across mini-batch samples, denoted by $\mathcal{D}_i$, where the gradient norm is*

$$\|\nabla\ell(\boldsymbol{\theta}_i; \mathcal{D}_i)\|_1 = \sum_{j=1}^{d}\left|\nabla_{\theta_i^{(j)}}\ell(\boldsymbol{\theta}_i; \mathcal{D}_i)\right| \quad and \quad \nabla_{\theta_i^{(j)}}\ell(\boldsymbol{\theta}_i; \mathcal{D}_i) = \frac{1}{|\mathcal{D}_i|}\sum_{n=1}^{|\mathcal{D}_i|}\nabla_{\theta_i^{(j)}}\ell(\boldsymbol{\theta}_i; \boldsymbol{x}_n, \boldsymbol{y}_n).$$

*Under some regularity conditions such that $\mathbb{E}\left[(\boldsymbol{\theta}_k - \hat{\boldsymbol{\theta}}^\star)^3\right] = o(\frac{1}{k})$, it holds*

$$\mathcal{L}(\hat{\boldsymbol{\theta}}^\star) \geq \mathbb{E}[\mathcal{L}(\boldsymbol{\theta}_k)] - \frac{1}{2}\mathbb{E}\left[\boldsymbol{\theta}_k^\top \boldsymbol{F}(\hat{\boldsymbol{\theta}}^\star)\boldsymbol{\theta}_k\right] - \eta\mu_k\|\boldsymbol{H}(\hat{\boldsymbol{\theta}}^\star)\hat{\boldsymbol{\theta}}^\star\|_\infty - \frac{1}{2}(\hat{\boldsymbol{\theta}}^\star - 2\boldsymbol{\theta}_0)^\top \boldsymbol{H}(\hat{\boldsymbol{\theta}}^\star)\hat{\boldsymbol{\theta}}^\star + o\left(\frac{1}{k}\right).$$

Theorem 2 presents a lower bound of the minimum achievable training loss. Similar to the upper bound depicted in Theorem 4 in Appendix C, this bound relies on the local curvature of the loss landscape and the sample gradient. More precisely, a reduced minimum achievable training loss is attainable when the expected absolute sample gradients $\mu_k$ and the FR norm $\mathbb{E}[\boldsymbol{\theta}_k^\top \boldsymbol{F}(\hat{\boldsymbol{\theta}}^\star)\boldsymbol{\theta}_k]$ are high, or the expected current training loss $\mathbb{E}[\mathcal{L}(\boldsymbol{\theta}_k)]$ is low.

As an approximation of the Fisher (Schraudolph, 2002), the empirical Fisher information matrix (EFIM) is used, which is defined as $\hat{\boldsymbol{F}}(\boldsymbol{\theta}) = \frac{1}{N}\sum_{n=1}^{N}\frac{\partial}{\partial\boldsymbol{\theta}}\ell(\boldsymbol{\theta}; \boldsymbol{x}_n, \boldsymbol{y}_n)\frac{\partial}{\partial\boldsymbol{\theta}}\ell(\boldsymbol{\theta}; \boldsymbol{x}_n, \boldsymbol{y}_n)^\top$.

### 3.4 GENERALIZATION ERROR

The decomposition of expected test loss (Eq. 3) underscores the significance of generalization error. The association of local flatness of the loss landscape with better generalization in DNNs has been widely embraced (Keskar et al., 2017; Wu et al., 2017; Yao et al., 2018; Liang et al., 2019). This view was initially proposed by Hochreiter & Schmidhuber (1997) by showing that flat minima, requiring less information to characterize, should outperform sharp minima in terms of generalization.

In assessing flatness, most measurements approximate the local curvature of the loss landscape, characterizing flat minima as those with reduced Hessian eigenvalues (Wu et al., 2017). As Hessians are very expensive to calculate, the sample gradient variance is used to measure local curvature properties. A rigorous discussion can be found in (Li et al., 2023, Section 3.2.2).

### 3.5 ZERO-SHOT METRIC

Inspired by the theoretical insights, we propose SiGeo, jointly considering the minimum achievable training loss (estimated by the FR norm, average of absolute gradient estimates, and current training loss) and generalization error (estimated by sample variance of gradients). As Fig. 1b illustrated, the expected loss is proportional to the generalization error and minimum achievable training loss. The generalization error can be approximated by the gradient variance as discussed in Section 3.4 and the minimum achievable training loss can be approximated by absolute sample gradients, FR norm and current training loss, as depicted in Theorem 2. With experiments of different combinations and weights, we consider the following proxy formulation.

**Definition 1.** *Let* $\boldsymbol{\theta}^{(m)}$ *denote the parameters of the* $m$-*th module, i.e.* $\boldsymbol{\theta}^{(m)} \subseteq \boldsymbol{\theta}$. *Let* $k$ *denote the number of batches used to compute SiGeo. Given a neural network with* $M$ *modules (containing trainable parameters), the* *zero-shot* *from* *information* *theory and* *geometry* *of loss landscape (SiGeo) is defined as follows:*

$$SiGeo = \sum_{m=1}^{M} \lambda_1 \log \left( \left\| \boldsymbol{\mu}_k^{(m)} \oslash \boldsymbol{\sigma}_k^{(m)} \right\|_1 \right) + \lambda_2 \log \left( \boldsymbol{\theta}_k^{\top} \hat{\boldsymbol{F}}(\boldsymbol{\theta}_k) \boldsymbol{\theta}_k \right) - \lambda_3 \ell(\boldsymbol{\theta}_k; \mathcal{D}_k) \quad (7)$$

*where* $\boldsymbol{\mu}_k^{(m)} = \frac{1}{k} \sum_{i=0}^{k} |\nabla_{\boldsymbol{\theta}^{(m)}} \ell(\boldsymbol{\theta}_i; \mathcal{D}_i)|$ *is the average of absolute gradient estimate with respect to* $\boldsymbol{\theta}^{(m)}$ *and* $\boldsymbol{\sigma}_k^{(m)} = \sqrt{\frac{1}{k} \sum_{i=0}^{k} \left( \nabla_{\boldsymbol{\theta}^{(m)}} \ell(\boldsymbol{\theta}_i; \mathcal{D}_i) - \frac{1}{k} \sum_{i=0}^{k} \nabla_{\boldsymbol{\theta}^{(m)}} \ell(\boldsymbol{\theta}_i; \mathcal{D}_i) \right)^2}$ *is the standard deviation of gradient across batches. See Appendix E for the implementation details of SiGeo.*

For the first term, we could consider a neuron-wise formulation $\lambda_1 \log \left( \|\boldsymbol{\mu}_k \oslash \boldsymbol{\sigma}_k\|_1 \right)$ as an alternative to the module-wise one, which is adapted from ZiCo. As depicted in Fig. 2, the second and third terms in Eq. 7 are important in utilizing the information accrued during the training process.

**Remark 2.** Based on our theoretical framework, various formulations for the zero-shot proxy can be chosen. In particular, when both $\lambda_2$ and $\lambda_3$ are set to zero, SiGeo simplifies to ZiCo. Likewise, if we allow for complete warming up of the supernet and fine-tuning of the subnet during the search, SiGeo becomes equivalent to one-shot NAS when $\lambda_1$ and $\lambda_2$ are set to zero. In practice, SiGeo reaches SOTA performance using just four batches (see Section 4). Thus, we compute SiGeo with only four input batches ($k = 4$); this makes SiGeo computationally efficient.

## 4 EXPERIMENTS

We conduct 5 sets of experiments: (1) Empirical validation of the theoretical findings (2) Evaluation of the SiGeo on zero-shot NAS benchmarks; (3) Evaluation of the SiGeo on NAS benchmarks under various warm-up levels; (4) Evaluation of SiGeo on click-through-rate (CTR) benchmark with various warm-up levels using the search space/policy and training setting from Zhang et al. (2023); (5) Evaluation of SiGeo on CIFAR-10 and CIFAR-100 under the zero-shot setting using the same search space, search policy, and training settings from Lin et al. (2021a). In addition, we have an ablation study to assess the efficacy of each key component of SiGeo in Appendix 4.5.

### 4.1 SETUP

In Experiment (1), our primary goal is to validate Theorems 2. Notably, the sample variance of gradients has been extensively studied by Li et al. (2023) and thus we focus on the FR norm, the

mean of absolute gradients and current training loss. To achieve this goal, we constructed two-layer MLP with rectified linear unit (ReLU) activation (MLP-ReLU) networks with varying hidden dimensions, ranging from 2 to 48 in increments of 2. Networks were trained on the MNIST dataset for 3 epochs using the SGD optimizer with a learning rate of $0.02$. The batch size was set as 128.

In Experiment (2), we compare SiGeo with other zero-shot proxies on three NAS benchmarks, including (i) NAS-Bench-101 (Ying et al., 2019), a cell-based search space with 423,624 architectures, (ii) NAS-Bench-201 (Dong & Yang, 2020), consisting of 15,625 architectures, and (iii) NAS-Bench-301 (Zela et al., 2021), a surrogate benchmark for the DARTS search space (Liu et al., 2018), with $10^{18}$ total architectures. Experiments are performed in the zero-shot setting.

In Experiment (3), we comprehensively compare our method against ZiCo under various warm-up levels on the same search spaces and tasks as used in Experiment (2).

In Experiment (4), we demonstrate empirical evaluations on three popular RecSys benchmarks for Click-Through Rates (CTR) prediction: Criteo[1], Avazu[2] and KDD Cup 2012[3]. To identify the optimal child subnet within the NASRec search space – encompassing NASRec Small and NASRec-Full (Zhang et al., 2023) – we employ the effective regularized evolution technique (Real et al., 2019). All three datasets are pre-processed in the same fashion as AutoCTR (Song et al., 2020). We conduct our experiments under three supernet warm-up levels: 0%, 1%, and 100%, corresponding to zero-shot, sub-one-shot and one-shot settings; see details in Appendix F.3.

In Experiment (5), we evaluate the compatibility of SiGeo on zero-shot NAS by utilizing the evaluation search algorithm and settings from ZiCo on CIFAR-10 and CIFAR-100 datasets (Krizhevsky et al., 2009). Results and experiment settings are provided in Appendix H.

## 4.2 EMPIRICAL JUSTIFICATION FOR THE THEORY

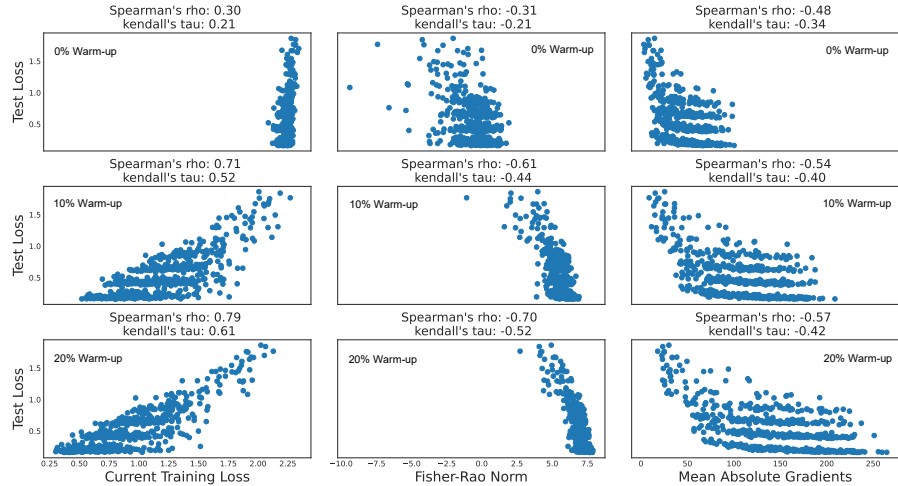

Figure 2: Test losses vs. statistics in Theorem 2. Results are generated by optimizing two-layer MLP-ReLU networks with varying hidden dimensions, ranging from 2 to 48 in increments of 2. The statistics for each network are computed after being warmed up with 0%, 10%, and 40% data.

Our empirical investigation starts with the validation of Theorem 2. The effectiveness of sample standard deviation of gradients (see Theorem 1) has been well validated by Li et al. (2023). Therefore, we focus on the correlations between training loss and test loss in relation to the other three key statistics: (i) current training loss, (ii) FR norm, and (iii) mean absolute gradients. Specifically, we conduct assessments on three different warm-up levels, i.e. 0%, 10% and 40%. The concrete training configures are described in Appendix F.1 and more results can be found in Appendix G.

Fig. 2 shows that networks with higher mean absolute gradients, FR norm values, or lower current training loss tend to have lower test and training loss. These results coincide with the conclusion drawn from Theorem 2. In addition, we observe a significant trend: as the warm-up level grows,

---

[1]https://www.kaggle.com/competitions/criteo-display-ad-challenge/data

[2]https://www.kaggle.com/competitions/avazu-ctr-prediction/data

[3]https://www.kaggle.com/competitions/kddcup2012-track2/data

there's a marked improvement in the ranking correlation between the current training loss and the FR norm with the final training/test loss. In contrast, the correlation of the mean absolute gradients remains fairly stable throughout this process. Given that the ZiCo proxy relies solely on the gradient standard deviation and the mean absolute gradients, the improved correlation coefficients of the FR norm and current training loss offer insights into why SiGeo outperforms the ZiCo method, particularly in the sub-one-shot setting.

### 4.3 VALIDATING SIGEO IN NAS BENCHMARKS

We assess the performance of SiGeo under various warm-up levels. This validation takes place across a range of well-established computer vision tasks. It's noteworthy that existing zero-shot methods predominantly focus their validation efforts on these tasks.

#### 4.3.1 COMPARING SIGEO WITH OTHER PROXIES UNDER THE ZERO-SHOT SETTING

We compute the correlation coefficients between proxies and test accuracy on several datasets. Specifically, we use the CIFAR10 (CF10) dataset from NASBench-101 (NB101), NASBench-201 (NB201), and NASBench-301 (NB301); the CIFAR100 (CF100) dataset from NB201; and the ImageNet16-120 (IMGNT) dataset from NB201. As presented in Table 2, SiGeo demonstrates either the highest or equally high correlation with the true test accuracy compared to other zero-shot proxies. Note that experiments are conducted without warm-up of candidate architectures.

Table 2: The correlation coefficients between various zero-cost proxies and two naive proxies (#params and FLOPs) vs. test accuracy on various NAS benchmkarks (Kendall and Spearman represent Kendall's $\tau$ and Spearman's $\rho$, respectively). The best results are shown with bold fonts.

| Proxy Name | NB101-CF10 | | NB201-CF10 | | NB201-CF100 | | NB201-IMGNT | | NB301-CF10 | |
|---|---|---|---|---|---|---|---|---|---|---|
| | Spearman | Kendall | Spearman | Kendall | Spearman | Kendall | Spearman | Kendall | Spearman | Kendall |
| epe-nas(Lopes et al., 2021) | 0.00 | 0.00 | 0.70 | 0.52 | 0.60 | 0.43 | 0.33 | 0.23 | 0.00 | 0.00 |
| fisher(Turner et al., 2020) | -0.28 | -0.20 | 0.50 | 0.37 | 0.54 | 0.40 | 0.48 | 0.36 | -0.28 | -0.19 |
| FLOPs (Ning et al., 2021) | 0.36 | 0.25 | 0.69 | 0.50 | 0.71 | 0.52 | 0.67 | 0.48 | 0.42 | 0.29 |
| grad-norm (Abdelfattah et al., 2021) | -0.25 | -0.17 | 0.58 | 0.42 | 0.63 | 0.47 | 0.57 | 0.42 | -0.04 | -0.03 |
| grasp(Wang et al., 2020a) | 0.27 | 0.18 | 0.51 | 0.35 | 0.54 | 0.38 | 0.55 | 0.39 | 0.34 | 0.23 |
| jacov (Mellor et al., 2021) | -0.29 | -0.20 | 0.75 | 0.57 | 0.71 | 0.54 | 0.71 | 0.54 | -0.04 | -0.03 |
| l2-norm (Abdelfattah et al., 2021) | 0.50 | 0.35 | 0.68 | 0.49 | 0.72 | 0.52 | 0.69 | 0.50 | 0.45 | 0.31 |
| NASWOT (Mellor et al., 2021) | 0.31 | 0.21 | 0.77 | 0.58 | 0.80 | 0.62 | 0.77 | 0.59 | 0.47 | 0.32 |
| #params (Ning et al., 2021) | 0.37 | 0.25 | 0.72 | 0.54 | 0.73 | 0.55 | 0.69 | 0.52 | 0.46 | 0.31 |
| plain (Abdelfattah et al., 2021) | -0.32 | -0.22 | -0.26 | -0.18 | -0.21 | -0.14 | -0.22 | -0.15 | -0.32 | -0.22 |
| snip (Lee et al., 2019) | -0.19 | -0.14 | 0.58 | 0.43 | 0.63 | 0.47 | 0.57 | 0.42 | -0.05 | -0.03 |
| synflow (Tanaka et al., 2020) | 0.31 | 0.21 | 0.73 | 0.54 | 0.76 | 0.57 | 0.75 | 0.56 | 0.18 | 0.12 |
| Zen (Lin et al., 2021b) | 0.59 | 0.42 | 0.35 | 0.27 | 0.35 | 0.28 | 0.39 | 0.29 | 0.43 | 0.30 |
| ZiCo (Li et al., 2023) | **0.63** | **0.46** | 0.74 | 0.54 | 0.78 | 0.58 | 0.79 | 0.60 | **0.50** | **0.35** |
| SiGeo (Ours) | **0.63** | **0.46** | **0.78** | **0.58** | **0.82** | **0.62** | **0.80** | **0.61** | **0.50** | **0.35** |

#### 4.3.2 COMPARING SIGEO WITH ZICO ON VARIOUS WARM-UP LEVELS

We conduct experiments to compare SiGeo against the SOTA proxy ZiCo under a sub-one-shot setting. Experiments are performed under the same setting of Section 4.3.1 with two key differences: (1) candidate architectures are warmed up before calculating the proxy scores; (2) we set $\lambda_2 = 50$ and $\lambda_3 = 1$ when the warm-up level is greater than zero. The results in Table 3 show (1) the ranking correlation of ZiCo does not improve much with more warm-up; (2) the ranking correlation of SiGeo improves significantly as the warm-up level increases. These results are consistent with the results in Section 4.2, underscoring the importance of the Fisher-Rao (FR) norm and current training loss in predicting the network performance when the candidate architectures are warmed up.

Table 3: The correlation coefficients of SiGeo and ZiCo vs. test accuracy on various warm-up levels.

| Benchmark | | NB101-CF10 | | NB201-CF10 | | NB201-CF100 | | NB201-IMGNT | | NB301-CF10 | |
|---|---|---|---|---|---|---|---|---|---|---|---|
| Method | Warm-up Level | Spearman | Kendall | Spearman | Kendall | Spearman | Kendall | Spearman | Kendall | Spearman | Kendall |
| ZiCo | 0% | 0.63 | 0.46 | 0.74 | 0.54 | 0.78 | 0.58 | 0.79 | 0.60 | 0.5 | 0.35 |
| ZiCo | 10% | 0.63 | 0.46 | 0.78 | 0.58 | 0.81 | 0.61 | 0.80 | 0.60 | 0.51 | 0.36 |
| ZiCo | 20% | 0.64 | 0.46 | 0.77 | 0.57 | 0.81 | 0.62 | 0.79 | 0.59 | 0.51 | 0.36 |
| ZiCo | 40% | 0.64 | 0.46 | 0.78 | 0.58 | 0.80 | 0.61 | 0.79 | 0.59 | 0.52 | 0.36 |
| SiGeo | 0% | 0.63 | 0.46 | 0.78 | 0.58 | 0.82 | 0.62 | 0.80 | 0.61 | 0.5 | 0.35 |
| SiGeo | 10% | 0.68 | 0.48 | 0.83 | 0.64 | 0.85 | 0.66 | 0.85 | 0.67 | 0.53 | 0.37 |
| SiGeo | 20% | 0.69 | 0.51 | 0.84 | 0.65 | 0.87 | 0.69 | 0.86 | 0.68 | 0.55 | 0.40 |
| SiGeo | 40% | 0.70 | 0.52 | 0.83 | 0.64 | 0.88 | 0.70 | 0.87 | 0.69 | 0.56 | 0.41 |

## 4.4 RECSYS BENCHMARK RESULTS

We use three RecSys benchmarks to validate the performance of SiGeo under different warm-up levels when compared with ZiCo and one-shot NAS approaches as well as hand-crafted models. Fig. 3 visualize the evaluation of our SiGeo-NAS against SOTA one-shot NAS baselines (DNAS(Krishna et al., 2021), PROFIT(Gao et al., 2021), AutoCTR(Song et al., 2020), NASRecNaet (Zhang et al., 2023)), and SOTA zero-shot NAS baseline (ZiCo (Li et al., 2023)) on three benchmark datasets: Criteo, Avazu, and KDD Cup 2012. All methods are trained in the NASRec-Full and NASRec-Small search spaces (Zhang et al., 2023). The **detailed result** can be found in Table 5 of Appendix I.

From Fig. 3, we observe that, after warming up the supernet using only 1% of the training data (sub-one-shot setting), SiGeo shows remarkable performance in comparison to established one-shot SOTA benchmarks with about 3X less computation time. In addition, when comparing SiGeo with hand-crafted CTR models (Guo et al., 2017; Lian et al., 2018; Naumov et al., 2019; Song et al., 2019) in Table 5, SiGeo-NAS demonstrates significantly improved performance.

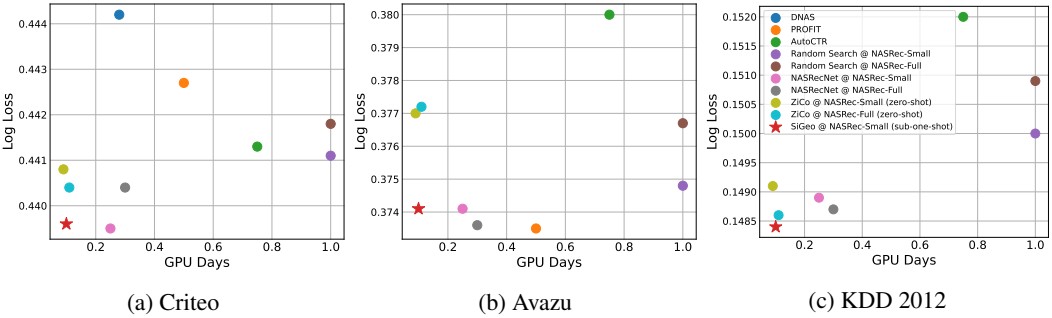

(a) Criteo        (b) Avazu        (c) KDD 2012

Figure 3: Performance of SiGeo-NAS on CTR Predictions Tasks. NASRec-Small and NASRec-Full are the two search spaces, and NASRecNet is the NAS method from Zhang et al. (2023).

## 4.5 ABLATION STUDY

To evaluate the effects of two key components, namely ZiCo and the FR norm, on SiGeo's performance, we carried out an ablation study. Specifically, we compare the best subnets identified using ZiCo and the FR norm as proxies. All experiments are performed in a sub-one-shot setting with 1% warm-up, following the same configuration as outlined in Section 4.4. The top-15 models selected by each proxy are trained from scratch and their test accuracies are illustrated as boxplots in Fig. 4. The results reveal that the exclusion of any terms from SiGeo detrimentally affects performance.

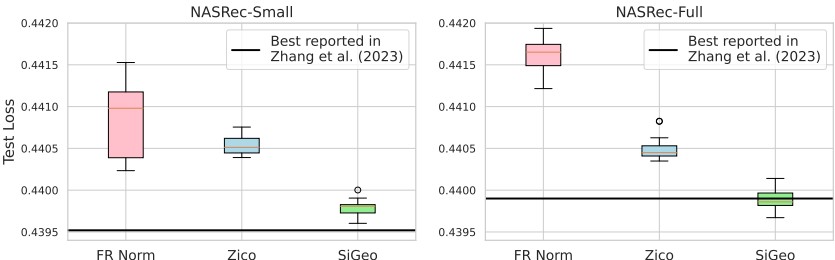

Figure 4: Evaluating the performance of two critical components (FR norm and ZiCo).

## 5 CONCLUSION

In this paper, we propose SiGeo, a new proxy proven to be increasingly effective when the candidate architectures continue to warm up. As the main theoretical contribution, we first present theoretical results that illuminate the connection between the minimum achievable test loss with the average of absolute gradient estimate, gradient standard deviation, FR norm, and the training loss under the sub-one-shot setting. Motivated by the theoretical insight, we have demonstrated that SiGeo achieves remarkable performance on three RecSys tasks (Criteo/Avazu/KDD-2012) with significantly lower search costs. In addition, we also validate SiGeo on various established NAS benchmarks (NASBench-101/NASBench-201/NASBench-301).

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

# A  ASSUMPTION JUSTIFICATION

The assumption **A.1** is originally proposed by Amari (1998) to show the Fisher efficiency result. This assumption has been widely used by many studies (Lee et al., 2022; Karakida et al., 2019; Thomas et al., 2020) to suggest that the model is powerful enough to capture the training distribution at $\boldsymbol{\theta} = \hat{\boldsymbol{\theta}}^\star$. We emphasize that Assumption **A.1** serves solely as a theoretical justification for substituting the Hessian matrix with the Fisher Information Matrix. Even in cases where this assumption fails to hold, the study by Thomas et al. (2020) has shown that the matrices tend to converge as the model training progresses.

The assumption for the initiation of supernet **A.2** holds importance in our theoretical analysis. This assumption aligns with recent theoretical findings for SGD. Specifically, after the warm-up phase, where the supernet is trained using a limited subset of the training data, the weights of the supernet are expected to escape the saddle points, settling near a specific local minimum (Fang et al., 2019; Kleinberg et al., 2018; Xu et al., 2018). In practice, this assumption is readily attainable: If **A.2** doesn't hold, one can simply introduce more samples to adequately warm up the supernet. In addition, we also observe that for numerous tasks, particularly those in the realm of computer vision, achieving satisfactory performance doesn't necessarily require warming up the supernet.

In most deep learning literature, the almost everywhere differentiability assumption **A.3** typically holds due to the need to calculate the gradient.

Lastly, **A.4** is the regularity condition that provides a foundational basis for performing mathematical operations that rely on the existence of the inverse of these matrices. In addition, this assumption directly implies the convexity of the loss function in the space $\mathbb{B}$.

# B  PROOF OF THEOREM 1

**Lemma 1.** *Assume A.4. If $\ell$ is differentiable, for any $(\boldsymbol{x}, \boldsymbol{y}) \in \mathcal{X} \times \mathcal{Y}$, it holds*

$$\ell(\boldsymbol{\theta}_1) \geq \ell(\boldsymbol{\theta}_2) + \langle \nabla \ell(\boldsymbol{\theta}_2; \boldsymbol{x}, \boldsymbol{y}), \boldsymbol{\theta}_1 - \boldsymbol{\theta}_2 \rangle, \text{ for all } \boldsymbol{\theta}_1, \boldsymbol{\theta}_2 \in \mathbb{B}$$

*Proof.* The positive definite Hessian in Assumption **A.4** directly implies the convexity of loss function $\ell$ with respect to $\boldsymbol{\theta}$ for all $(\boldsymbol{x}, \boldsymbol{y}) \in \mathcal{X} \times \mathcal{Y}$. Then the conclusion immediately follow Garrigos & Gower (2023, Lemma 2.8). □

**Theorem** 1. *Assume A.1-A.4. Consider $\{\boldsymbol{\theta}_k\}_{k \in \mathbb{Z}}$ a sequence generated by the first-order (SGD) algorithm (4), with a decreasing sequence of stepsizes satisfying $\eta_k > 0$. Let $\sigma_k^2 := \mathrm{Var}[\nabla \ell(\boldsymbol{\theta}_k; \mathcal{D}_k)]$ denote the variance of sample gradient, where the variance is taken over the joint distribution of all the training samples until the $k$-th iteration (also known as filtration in stochastic approximation literature). Then It holds that*

$$\mathbb{E}[\mathcal{L}(\bar{\boldsymbol{\theta}}_k)] - \mathcal{L}(\hat{\boldsymbol{\theta}}^\star) \leq \frac{\|\boldsymbol{\theta}_0 - \hat{\boldsymbol{\theta}}^\star\|}{2 \sum_{t=0}^{k-1} \eta_t} + \frac{1}{2} \sum_{i=0}^{k-1} \sigma_k^2$$

*Proof.* For a given weight $\boldsymbol{\theta}_k$, the sample gradient estimate is an unbiased estimate of expected gradient, i.e. $\mathbb{E}[\nabla \ell(\boldsymbol{\theta}_k; \mathcal{D}_k)|\boldsymbol{\theta}_k] = \mathbb{E}[\frac{1}{|\mathcal{D}_k|} \sum_{(\boldsymbol{x}_n, \boldsymbol{y}_n) \in \mathcal{D}_k} \nabla \ell(\boldsymbol{\theta}_k; \boldsymbol{x}_n, \boldsymbol{y}_n)|\boldsymbol{\theta}_k] = \nabla \mathcal{L}(\boldsymbol{\theta}_k)$ under some mild conditions. We will note $\mathbb{E}_k[\cdot]$ instead of $\mathbb{E}[\cdot|\boldsymbol{\theta}_k]$, and $\mathrm{Var}_k[\cdot]$ instead of $\mathrm{Var}[\cdot|\boldsymbol{\theta}_k]$, for simplicity. Thus, we have the gradient variance

$$\mathrm{Var}_k[\nabla \ell(\boldsymbol{\theta}_k; \mathcal{D}_k)] = \mathbb{E}_k\left[\|\nabla \ell(\boldsymbol{\theta}_k; \mathcal{D}_k)\|^2\right] - \|\mathbb{E}_k\left[\nabla \ell(\boldsymbol{\theta}_k; \mathcal{D}_k)\right]\|^2 = \mathbb{E}_k\left[\|\nabla \ell(\boldsymbol{\theta}_k; \mathcal{D}_k)\|^2\right]. \quad (8)$$

Let us start by analyzing the behavior of $\|\boldsymbol{\theta}_k - \hat{\boldsymbol{\theta}}^\star\|^2$. By expanding the squares, we obtain

$$\|\boldsymbol{\theta}_{k+1} - \hat{\boldsymbol{\theta}}^\star\| = \|\boldsymbol{\theta}_k - \boldsymbol{\theta}^\star\| - 2\eta_k \langle \nabla \ell(\boldsymbol{\theta}_k; \mathcal{D}_k), \boldsymbol{\theta}_k - \boldsymbol{\theta}^\star \rangle + \eta_k^2 \|\nabla \ell(\boldsymbol{\theta}_k; \mathcal{D}_k)\|^2$$

Hence, after taking the expectation conditioned on $\boldsymbol{\theta}_k$, we can use the convexity of $\ell$ to obtain:

$$
\begin{aligned}
\mathbb{E}_k \left[ \|\boldsymbol{\theta}_{k+1} - \hat{\boldsymbol{\theta}}^\star\| \right] &= \|\boldsymbol{\theta}_k - \hat{\boldsymbol{\theta}}^\star\|^2 - 2\eta_k \langle \mathbb{E}_k[\nabla\ell(\boldsymbol{\theta}_k; \mathcal{D}_k)], \boldsymbol{\theta}_k - \boldsymbol{\theta}^\star \rangle + \eta_k^2 \mathbb{E}_k \left[ \|\nabla\ell(\boldsymbol{\theta}_k; \mathcal{D}_k)\|^2 \right] \\
&\leq \|\boldsymbol{\theta}_k - \hat{\boldsymbol{\theta}}^\star\|^2 + 2\eta_k \langle \nabla\mathcal{L}(\boldsymbol{\theta}_k), \boldsymbol{\theta}^\star - \boldsymbol{\theta}_k \rangle + \eta_k^2 \mathbb{E}_k \left[ \|\nabla\ell(\boldsymbol{\theta}_k; \mathcal{D}_k)\|^2 \right] \\
&\leq \|\boldsymbol{\theta}_k - \hat{\boldsymbol{\theta}}^\star\|^2 + 2\eta_k \left( \mathcal{L}(\boldsymbol{\theta}^\star) - \mathcal{L}(\boldsymbol{\theta}_k) \right) + \eta_k^2 \mathrm{Var}_k \left[ \nabla\ell(\boldsymbol{\theta}_k; \mathcal{D}_k) \right] \quad (9)
\end{aligned}
$$

where the last equation holds due to Lemma 1 and Eq. (8). Rearranging and taking the full expectation of Eq. (9) over all past training samples until iteration $k$, we have

$$
2\eta_k \left( \mathbb{E}\left[\mathcal{L}(\boldsymbol{\theta}_k)\right] - \mathcal{L}(\boldsymbol{\theta}^\star) \right) \leq \mathbb{E}\left[ \|\boldsymbol{\theta}_k - \hat{\boldsymbol{\theta}}^\star\| \right] - \mathbb{E}\left[ \|\boldsymbol{\theta}_{k+1} - \hat{\boldsymbol{\theta}}^\star\| \right] + \eta_k^2 \mathbb{E}\left[ \mathrm{Var}_k \left[ \nabla\ell(\boldsymbol{\theta}_k; \mathcal{D}_k) \right] \right] \quad (10)
$$

Due to law of total variance, it holds

$$
\sigma^2 = \mathrm{Var}[\nabla\ell(\boldsymbol{\theta}_k; \mathcal{D}_k)] = \mathbb{E}\left[ \mathrm{Var}\left[ \nabla\ell(\boldsymbol{\theta}_k; \mathcal{D}_k|\boldsymbol{\theta}_k) \right] \right] + \mathrm{Var}\left[ \mathbb{E}\left[ \nabla\ell(\boldsymbol{\theta}_k; \mathcal{D}_k)|\boldsymbol{\theta}_k \right] \right] \geq \mathbb{E}\left[ \mathrm{Var}_k \left[ \nabla\ell(\boldsymbol{\theta}_k; \mathcal{D}_k) \right] \right].
$$

Then Eq. 10 becomes

$$
2\eta_k \left( \mathbb{E}\left[\mathcal{L}(\boldsymbol{\theta}_k)\right] - \mathcal{L}(\boldsymbol{\theta}^\star) \right) \leq \mathbb{E}\left[ \|\boldsymbol{\theta}_k - \hat{\boldsymbol{\theta}}^\star\| \right] - \mathbb{E}\left[ \|\boldsymbol{\theta}_{k+1} - \hat{\boldsymbol{\theta}}^\star\| \right] + \eta_k^2 \sigma^2
$$

Summing over $i = 0, 1, \ldots, k-1$ and using telescopic cancellation gives

$$
2 \sum_{i=0}^{k-1} \eta_i \left( \mathbb{E}\left[\mathcal{L}(\boldsymbol{\theta}_i)\right] - \mathcal{L}(\boldsymbol{\theta}^\star) \right) \leq \|\boldsymbol{\theta}_0 - \hat{\boldsymbol{\theta}}^\star\| - \mathbb{E}\left[ \|\boldsymbol{\theta}_k - \hat{\boldsymbol{\theta}}^\star\| \right] + \sum_{i=0}^{k} \eta_i^2 \sigma^2
$$

Since $\mathbb{E}\left[ \|\boldsymbol{\theta}_k - \hat{\boldsymbol{\theta}}^\star\| \right] \geq 0$, dividing both sides by $2\sum_{t=0}^{k-1} \eta_t$ gives:

$$
\sum_{i=0}^{k-1} \frac{\eta_i}{\sum_{t=0}^{k-1} \eta_t} \left( \mathbb{E}\left[\mathcal{L}(\boldsymbol{\theta}_i)\right] - \mathcal{L}(\boldsymbol{\theta}^\star) \right) \leq \frac{\|\boldsymbol{\theta}_0 - \hat{\boldsymbol{\theta}}^\star\|}{2\sum_{t=0}^{k-1} \eta_t} + \sum_{i=0}^{k} \frac{\eta_i^2}{2\sum_{t=0}^{k-1} \eta_t} \sigma^2
$$

Define for $i = 0, 1, \ldots, k-1$

$$
p_{k,i} := \frac{\eta_i^2}{\sum_{t=0}^{k-1} \eta_t}
$$

and observe that $p_{k,i} \geq 0$ and $\sum_{i=0}^{k-1} p_{k,i} = 1$. This allows us to treat the $\{p_{k,i}\}$ as probabilities. Then using the fact that $\ell$ is convex together with Jensen's inequality gives

$$
\begin{aligned}
\mathbb{E}[\mathcal{L}(\bar{\boldsymbol{\theta}}_k)] - \mathcal{L}(\hat{\boldsymbol{\theta}}^\star) &\leq \sum_{i=0}^{k-1} \frac{\eta_i}{\sum_{t=0}^{k-1} \eta_t} \left( \mathbb{E}\left[\mathcal{L}(\boldsymbol{\theta}_i)\right] - \mathcal{L}(\boldsymbol{\theta}^\star) \right) \\
&\leq \frac{\|\boldsymbol{\theta}_0 - \hat{\boldsymbol{\theta}}^\star\|}{2\sum_{t=0}^{k-1} \eta_t} + \sum_{i=0}^{k} \frac{\eta_i^2}{2\sum_{t=0}^{k-1} \eta_t} \mathrm{Var}\left[ \nabla\ell(\boldsymbol{\theta}_i; \mathcal{D}_i) \right] \\
&\leq \frac{\|\boldsymbol{\theta}_0 - \hat{\boldsymbol{\theta}}^\star\|}{2\sum_{t=0}^{k-1} \eta_t} + \frac{1}{2} \sum_{i=0}^{k-1} \sigma_k^2 \quad (11)
\end{aligned}
$$

where the last inequality holds because $p_{k,i} \leq 1$. $\qquad \square$

## C  CONVERGENCE ANALYSIS WITH STRONG CONVEXITY ASSUMPTION

In this section, we will extend our analysis to strong convex loss function, a stronger assumption than convexity. Mathematically, we assume that the expected loss function of candidate network $\mathcal{L}(\boldsymbol{\theta})$ is locally strongly convex with convexity constant $M$, that is, $\mathcal{L}(\boldsymbol{\theta}_1) \geq \mathcal{L}(\boldsymbol{\theta}_2) + \langle \nabla\mathcal{L}(\boldsymbol{\theta}_2), \boldsymbol{\theta}_1 - \boldsymbol{\theta}_2 \rangle + \frac{M}{2}\|\boldsymbol{\theta}_1 - \boldsymbol{\theta}_2\|^2$ for any $\boldsymbol{\theta}_1, \boldsymbol{\theta}_2 \in \mathbb{B}$.

In addition, we define the covariance matrix of gradient

$$
\boldsymbol{C} = \mathbb{E}\left[ \left( \nabla\ell(\hat{\boldsymbol{\theta}}^\star; \boldsymbol{x}, \boldsymbol{y}) - \mathbb{E}[\nabla\ell(\hat{\boldsymbol{\theta}}^\star; \boldsymbol{x}, \boldsymbol{y})] \right) \left( \nabla\ell(\hat{\boldsymbol{\theta}}^\star; \boldsymbol{x}, \boldsymbol{y}) - \mathbb{E}[\nabla\ell(\hat{\boldsymbol{\theta}}^\star; \boldsymbol{x}, \boldsymbol{y})] \right)^\top \right].
$$

Subsequently, the variance of sample gradient is denoted by $\mathrm{Tr}(\boldsymbol{C}) = \mathrm{Var}[\nabla\ell(\hat{\boldsymbol{\theta}}^{\star}; \boldsymbol{x}, \boldsymbol{y})]$.

Based on the strong convexity assumption, the first theorem is established by using a result from Moulines & Bach (2011); see the proof in Appendix C.1.

**Theorem 3.** *Assume A.1-A.4. If the loss function is strongly convex, under the regularity conditions in Moulines & Bach (2011), for the average iterate $\bar{\boldsymbol{\theta}}_k = \frac{1}{k+1}\sum_{i=0}^{k}\boldsymbol{\theta}_i$ with a first-order SGD (4) ($\boldsymbol{B}_k = \boldsymbol{I}$), it holds*

$$\mathbb{E}[\ell(\bar{\boldsymbol{\theta}}_k; \boldsymbol{x}, \boldsymbol{y}) - \ell(\hat{\boldsymbol{\theta}}^{\star}; \boldsymbol{x}, \boldsymbol{y})] \leq L\mathbb{E}\left[\|\bar{\boldsymbol{\theta}}_k - \hat{\boldsymbol{\theta}}^{\star}\|^2\right]^{1/2} \leq \mathcal{O}\left(\frac{\mathrm{Tr}\left(\boldsymbol{F}(\hat{\boldsymbol{\theta}}^{\star})^{-1}\right)\sqrt{\mathrm{Tr}(\boldsymbol{C})}}{\sqrt{k+1}}\right) + o\left(\frac{1}{\sqrt{k+1}}\right)$$

Then we present our main theorem which relates the optimality gap with FR norm and standard deviation values. The proof of the following theorem can be found in Appendix C.2.

**Theorem 4.** *Assume A.1-A.4. Consider the first order SGD update 4 ($\boldsymbol{B}_k = \boldsymbol{I}$) with fixed learning rate $\eta$. If the loss function is strongly convex, under the same assumption as stated in Lemma 3, the expected loss is bounded as follows*

$$\mathbb{E}[\ell(\bar{\boldsymbol{\theta}}_k; \boldsymbol{x}, \boldsymbol{y})] - \mathbb{E}[\ell(\hat{\boldsymbol{\theta}}^{\star}; \boldsymbol{x}, \boldsymbol{y})]$$
$$\leq \mathcal{O}\left(\frac{\sqrt{\mathrm{Tr}(\boldsymbol{C})}}{2\sqrt{k+1}}\left[\frac{d^2 M}{\mathbb{E}[\boldsymbol{\theta}_k \boldsymbol{F}(\hat{\boldsymbol{\theta}}^{\star})\boldsymbol{\theta}_k] + 2\eta\mathbb{E}[\sum_{i=0}^{k}\nabla\ell(\boldsymbol{\theta}_i; \mathcal{D}_i)]^{\top}\boldsymbol{F}(\hat{\boldsymbol{\theta}}^{\star})\hat{\boldsymbol{\theta}}^{\star} + (\boldsymbol{\theta}^{\star} - 2\boldsymbol{\theta}_0)^{\top}\boldsymbol{F}(\hat{\boldsymbol{\theta}}^{\star})\hat{\boldsymbol{\theta}}^{\star}} + \frac{d\lambda_1(\boldsymbol{F}(\hat{\boldsymbol{\theta}}^{\star})) - d}{\lambda_d(\boldsymbol{F}(\hat{\boldsymbol{\theta}}^{\star}))}\right]\right)$$

*where the big O notation hides the maximum and minimum eigenvalue of Fisher's information.*

Theorem 4 provides insights about the (local) optimality gap, which is tied to the FR norm $\mathbb{E}[\boldsymbol{\theta}_k^{\top}\boldsymbol{F}(\hat{\boldsymbol{\theta}}^{\star})\boldsymbol{\theta}_k]$, and the gradient variance $\mathrm{Tr}(\boldsymbol{C})$. In other words, the higher the FR norm or the smaller the gradient variance across different training samples, the lower the training loss the model converges to; i.e., the network converges at a faster rate.

## C.1 PROOF OF THEOREM 3

**Lemma 2.** *Assume A.3. The loss function of candidate network $\ell(\boldsymbol{\theta}; \boldsymbol{x}, \boldsymbol{y})$ is locally Lipschitz continuous with a Lipschitz constant $L$ for any $(\boldsymbol{x}, \boldsymbol{y})$, that is, $|\ell(\boldsymbol{\theta}_1; \boldsymbol{x}, \boldsymbol{y}) - \ell(\boldsymbol{\theta}_2; \boldsymbol{x}, \boldsymbol{y})| \leq L\|\boldsymbol{\theta}_1 - \boldsymbol{\theta}_2\|$ for all $\boldsymbol{\theta}_1, \boldsymbol{\theta}_2 \in \mathbb{B}$.*

*Proof.* The differentiability of the loss $\nabla\ell(\boldsymbol{\theta}; \boldsymbol{x}, \boldsymbol{y})$ (Assumption **A.3**) implies the boundedness of $\nabla\ell$ on the compact set $\mathbb{B}$, i.e. $\|\nabla\ell(\boldsymbol{\theta}; \boldsymbol{x}, \boldsymbol{y})\| \leq L$ for all $\boldsymbol{x} \in \mathcal{X}, \boldsymbol{y} \in \mathcal{Y}$ and $\boldsymbol{\theta} \in \mathbb{B}$. Here, $L$ is a constant that bounds the gradient norm. The result can be obtained directly from the Mean Value Theorem (MVT) that

$$\|\ell(\boldsymbol{\theta}_1, \boldsymbol{x}, \boldsymbol{y}) - \ell(\boldsymbol{\theta}_2, \boldsymbol{x}, \boldsymbol{y})\| \leq \|\nabla\ell(\xi\boldsymbol{\theta}_1 + (1 - \xi)\boldsymbol{\theta}_2)\|\|\boldsymbol{\theta}_1 - \boldsymbol{\theta}_2\| \leq L_{\ell}\|\boldsymbol{\theta}_1 - \boldsymbol{\theta}_2\|$$

where $\xi \in (0, 1)$ is some constant. $\qquad\square$

**Lemma 3** (Theorem 3 Moulines & Bach (2011)). *Under some regularity conditions, for a first-order stochastic gradient descent (4) ($\boldsymbol{B}_k = \boldsymbol{I}$), we have:*

$$\mathbb{E}\left[\|\bar{\boldsymbol{\theta}}_k - \hat{\boldsymbol{\theta}}^{\star}\|^2\right] \leq \frac{\mathrm{Tr}\left(\boldsymbol{H}(\hat{\boldsymbol{\theta}}^{\star})^{-1}\boldsymbol{C}\boldsymbol{H}(\hat{\boldsymbol{\theta}}^{\star})^{-1}\right)}{k+1} + o\left(\frac{1}{k}\right)$$

*where the big O notation hides constants such as the upper bound of the norm of the Hessian matrix, strong convexity constant, and Lipschitz constant. We refer to Moulines & Bach (2011) for detailed information about regularity conditions.*

**Lemma 4.** *(Ruhe's trace inequality Ruhe (1970)). If $\boldsymbol{A}$ and $\boldsymbol{B}$ are positive semidefinite Hermitian matrices with eigenvalues,*

$$a_1 \geq a_2 \geq \ldots a_d \geq 0 \qquad b_1 \geq b_2 \geq \ldots b_d \geq 0$$

*respectively, then*

$$\mathrm{Tr}(\boldsymbol{A}\boldsymbol{B}) \leq \sum_{i=1}^{d} a_i b_i$$

**Lemma 5.** *Let $\boldsymbol{A}$ and $\boldsymbol{B}$ be two positive semipositive Hermitian matrix with proper dimension for multiplication. We have $\mathrm{Tr}(\boldsymbol{AB}) \leq \mathrm{Tr}(\boldsymbol{A})\,\mathrm{Tr}(\boldsymbol{B})$.*

*Proof.* Notice that positive semidefinite implies that all eigenvalues of matrix $\boldsymbol{A}$ and $\boldsymbol{B}$ are positive. Following Lemma 4, we have

$$\mathrm{Tr}(\boldsymbol{AB}) \leq \sum_{i=1}^{d} a_i b_i \overset{(*)}{\leq} \sum_{i=1}^{d} a_i \sum_{i=1}^{d} b_i = \mathrm{Tr}(\boldsymbol{A})\,\mathrm{Tr}(\boldsymbol{B})$$

where $(*)$ holds due to the fact that $\sum_{i=1}^{d} a_i \sum_i b_i = \sum_{i=1}^{d} a_i b_i + \sum_{i \neq j} a_i b_j \geq \sum_{i=1}^{d} a_i b_i$ for $a_i, b_i \geq 0$. $\qquad\square$

Theorem 3 immediately follows Lemma 3.

**Theorem** 3. *Assume A.1-A.4. If the loss function is strongly convex, under the regularity conditions in Moulines & Bach (2011), for the average iterate $\bar{\boldsymbol{\theta}}_k = \frac{1}{k+1}\sum_{i=0}^{k} \boldsymbol{\theta}_i$ with a first-order SGD (4) ($\boldsymbol{B}_k = \boldsymbol{I}$), it holds*

$$\mathbb{E}[\ell(\bar{\boldsymbol{\theta}}_k; \boldsymbol{x}, \boldsymbol{y}) - \ell(\hat{\boldsymbol{\theta}}^\star; \boldsymbol{x}, \boldsymbol{y})] \leq L\mathbb{E}\left[\|\bar{\boldsymbol{\theta}}_k - \hat{\boldsymbol{\theta}}^\star\|^2\right]^{1/2} \leq \mathcal{O}\left(\frac{\mathrm{Tr}\left(\boldsymbol{F}(\hat{\boldsymbol{\theta}}^\star)^{-1}\right)\sqrt{\mathrm{Tr}(\boldsymbol{C})}}{\sqrt{k}}\right) + o\left(\frac{1}{\sqrt{k}}\right)$$

*Proof.* The Lipschitz continuity of the loss function (Lemma 2) with a Lipschitz constant $L < \infty$ implies that

$$\mathbb{E}\left[\ell(\bar{\boldsymbol{\theta}}_k) - \ell(\hat{\boldsymbol{\theta}}^\star)\right] \leq L\mathbb{E}\left[\|\bar{\boldsymbol{\theta}}_k - \hat{\boldsymbol{\theta}}^\star\|\right] \leq L\mathbb{E}\left[\|\bar{\boldsymbol{\theta}}_k - \hat{\boldsymbol{\theta}}^\star\|^2\right]^{\frac{1}{2}}$$

Since $\boldsymbol{H}$ and $\boldsymbol{C}$ are both positive definite, applying Lemma 5 twice leads to

$$\mathrm{Tr}\left(\boldsymbol{H}(\hat{\boldsymbol{\theta}}^\star)^{-1}\boldsymbol{C}\boldsymbol{H}(\hat{\boldsymbol{\theta}}^\star)^{-1}\right) \leq \mathrm{Tr}\left(\boldsymbol{H}(\hat{\boldsymbol{\theta}}^\star)^{-1}\right)^2 \mathrm{Tr}(\boldsymbol{C}). \tag{12}$$

Using Lemma 3 and Eq. 12, we have

$$\mathbb{E}\left[\ell(\bar{\boldsymbol{\theta}}_k) - \ell(\hat{\boldsymbol{\theta}}^\star)\right]^2 \leq \frac{L^2 \mathrm{Tr}\left(\boldsymbol{H}(\hat{\boldsymbol{\theta}}^\star)^{-1}\boldsymbol{C}\boldsymbol{H}(\hat{\boldsymbol{\theta}}^\star)^{-1}\right)}{k+1} + o\left(\frac{1}{k}\right) \leq \frac{L^2 \mathrm{Tr}\left(\boldsymbol{H}(\hat{\boldsymbol{\theta}}^\star)^{-1}\right)^2 \mathrm{Tr}(\boldsymbol{C})}{k+1} + o\left(\frac{1}{k}\right)$$

Then the conclusion follows the inequality $\sqrt{a+b} \leq \sqrt{a} + \sqrt{b}$ for $a, b \geq 0$ and the "realizability" assumption **A.1**. $\qquad\square$

### C.2 PROOF OF THEOREM 4

To begin with, we present several technical lemmas to support the proof of Theorem 4.

**Lemma 6** (Theorem 1 Bai & Golub (1996)). *Let $\boldsymbol{A}$ be an d-by-d symmetric positive definite matrix, $\mu_1 = \mathrm{Tr}(\boldsymbol{A})$, $\mu_2 = \|\boldsymbol{A}\|_F^2$ and $\alpha \leq \lambda_d(\boldsymbol{A}) \leq \ldots \leq \lambda_1(\boldsymbol{A}) \leq \beta$ with $\alpha > 0$, then*

$$(\mu_1 \quad d)\begin{pmatrix} \mu_2 & \mu_1 \\ \beta^2 & \beta \end{pmatrix}^{-1}(d \quad 1) \leq \mathrm{Tr}(\boldsymbol{A}^{-1}) \leq (\mu_1 \quad d)\begin{pmatrix} \mu_2 & \mu_1 \\ \alpha^2 & \alpha \end{pmatrix}^{-1}(d \quad 1) \tag{13}$$

**Lemma 7.** *Suppose that $\alpha \leq \lambda_1(\boldsymbol{F}) \leq \beta$. Then for a d-by-d symmetric positive definite matrix $\boldsymbol{F}$, it holds that*

$$\mathrm{Tr}(\boldsymbol{F}^{-1}) \leq d^2 \mathrm{Tr}(\boldsymbol{F})^{-1} + \frac{d\beta}{\alpha} - \frac{d}{\alpha} \tag{14}$$

*Proof.* Applying Lemma 6 to $\mathrm{Tr}(\boldsymbol{F}^{-1})$ leads to

$$\mathrm{Tr}(\boldsymbol{F}^{-1}) \leq (\mathrm{Tr}(\boldsymbol{F}) \quad d)\begin{pmatrix} \|\boldsymbol{F}\|_F^2 & \mathrm{Tr}(\boldsymbol{F}) \\ \alpha^2 & \alpha \end{pmatrix}^{-1}(d \quad 1) = \frac{d\alpha \cdot \mathrm{Tr}(\boldsymbol{F}) - d^2\alpha^2 - \mathrm{Tr}(\boldsymbol{F})^2 + d\|\boldsymbol{F}\|_F^2}{\alpha\|\boldsymbol{F}\|_F^2 - \alpha^2 \mathrm{Tr}(\boldsymbol{F})} \tag{15}$$

where the last equality holds due to the definition of matrix inverse. The Eq. (15) can be further rearranged to

$$\text{Tr}(\boldsymbol{F}^{-1}) \leq -\frac{d}{\alpha} + \frac{d^2\alpha^2 + \text{Tr}(\boldsymbol{F})^2}{\alpha^2 \text{Tr}(\boldsymbol{F}) - \alpha\|\boldsymbol{F}\|_F^2} \leq -\frac{d}{\alpha} + \frac{d^2\alpha^2 + \text{Tr}(\boldsymbol{F})^2}{\alpha^2 \text{Tr}(\boldsymbol{F})} \leq -\frac{d}{\alpha} + \frac{d^2}{\text{Tr}(\boldsymbol{F})} + \frac{\text{Tr}(\boldsymbol{F})}{\alpha} \quad (16)$$

The conclusion (14) can be obtained by noting that $\text{Tr}(\boldsymbol{F}) = \sum_{i=1}^{d} \lambda_i(\boldsymbol{F}) \leq d\beta$. $\qquad\square$

**Theorem** 4 *Assume A.1-A.4. Consider the first order SGD update 4 ($\boldsymbol{B}_k = \boldsymbol{I}$) with fixed learning rate $\eta$. If the loss function is strongly convex, under the same assumption as stated in Lemma 3, the expected loss is bounded as follows*

$$\mathbb{E}[\ell(\bar{\boldsymbol{\theta}}_k; \boldsymbol{x}, \boldsymbol{y})] - \mathbb{E}[\ell(\hat{\boldsymbol{\theta}}^{\star}; \boldsymbol{x}, \boldsymbol{y})]$$
$$\leq \mathcal{O}\left(\frac{\sqrt{\text{Tr}(\boldsymbol{C})}}{2\sqrt{k+1}}\left[\frac{d^2 M}{\mathbb{E}\left[\boldsymbol{\theta}_k \boldsymbol{F}(\hat{\boldsymbol{\theta}}^{\star})\boldsymbol{\theta}_k\right] + 2\eta\mathbb{E}\left[\sum_{i=0}^{k}\nabla\ell(\boldsymbol{\theta}_i;\mathcal{D}_i)\right]^{\top}\boldsymbol{F}(\hat{\boldsymbol{\theta}}^{\star})\hat{\boldsymbol{\theta}}^{\star} + (\boldsymbol{\theta}^{\star}-2\boldsymbol{\theta}_0)^{\top}\boldsymbol{F}(\hat{\boldsymbol{\theta}}^{\star})\hat{\boldsymbol{\theta}}^{\star}} + \frac{d\lambda_1(\boldsymbol{F}(\hat{\boldsymbol{\theta}}^{\star}))-d}{\lambda_d(\boldsymbol{F}(\hat{\boldsymbol{\theta}}^{\star}))}\right]\right)$$

*where the big O notation hides the maximum and minimum eigenvalue of Fisher's information.*

*Proof.* To begin with, we have

$$\text{Tr}\left(\boldsymbol{F}(\hat{\boldsymbol{\theta}}^{\star})^{-1}\right) \leq \frac{d^2}{\text{Tr}(\boldsymbol{F}(\hat{\boldsymbol{\theta}}^{\star}))} + \frac{d\lambda_1(\boldsymbol{F}(\hat{\boldsymbol{\theta}}^{\star})) - d}{\lambda_d(\boldsymbol{F}(\hat{\boldsymbol{\theta}}^{\star}))} \quad (17)$$

where the inequality holds because of Lemma 7 and choosing $\alpha = \lambda_d(\boldsymbol{F})$ and $\beta = \lambda_1(\boldsymbol{F})$. Further, note that

$$\mathbb{E}\left[\text{Tr}\left((\boldsymbol{\theta}_k - \hat{\boldsymbol{\theta}}^{\star})^{\top}\boldsymbol{F}(\hat{\boldsymbol{\theta}}^{\star})(\boldsymbol{\theta}_k - \hat{\boldsymbol{\theta}}^{\star})\right)\right] = \mathbb{E}\left[\text{Tr}\left(\boldsymbol{F}(\hat{\boldsymbol{\theta}}^{\star})(\boldsymbol{\theta}_k - \hat{\boldsymbol{\theta}}^{\star})(\boldsymbol{\theta}_k - \hat{\boldsymbol{\theta}}^{\star})^{\top}\right)\right]$$
$$\leq \text{Tr}\left(\boldsymbol{F}(\hat{\boldsymbol{\theta}}^{\star})\right)\mathbb{E}\left[\text{Tr}\left((\boldsymbol{\theta}_k - \hat{\boldsymbol{\theta}}^{\star})(\boldsymbol{\theta}_k - \hat{\boldsymbol{\theta}}^{\star})^{\top}\right)\right] \quad (18)$$

Notice that $\text{Tr}\left((\boldsymbol{\theta}_k - \hat{\boldsymbol{\theta}}^{\star})(\boldsymbol{\theta}_k - \hat{\boldsymbol{\theta}}^{\star})^{\top}\right) = \|\boldsymbol{\theta}_k - \hat{\boldsymbol{\theta}}^{\star}\|^2$. Then Eq. 18 becomes

$$\frac{1}{\text{Tr}\left(\boldsymbol{F}(\hat{\boldsymbol{\theta}}^{\star})\right)} \leq \frac{\mathbb{E}\left[\|\boldsymbol{\theta}_k - \hat{\boldsymbol{\theta}}^{\star}\|^2\right]}{\mathbb{E}\left[\text{Tr}\left((\boldsymbol{\theta}_k - \hat{\boldsymbol{\theta}}^{\star})^{\top}\boldsymbol{F}(\hat{\boldsymbol{\theta}}^{\star})(\boldsymbol{\theta}_k - \hat{\boldsymbol{\theta}}^{\star})\right)\right]}$$
$$= \frac{\mathbb{E}\left[\|\boldsymbol{\theta}_k - \hat{\boldsymbol{\theta}}^{\star}\|^2\right]}{\mathbb{E}\left[\boldsymbol{\theta}_k \boldsymbol{F}(\hat{\boldsymbol{\theta}}^{\star})\boldsymbol{\theta}_k\right] - 2\mathbb{E}\left[\boldsymbol{\theta}_k\right]^{\top}\boldsymbol{F}(\hat{\boldsymbol{\theta}}^{\star})\hat{\boldsymbol{\theta}}^{\star} + \boldsymbol{\theta}^{\star\top}\boldsymbol{F}(\hat{\boldsymbol{\theta}}^{\star})\hat{\boldsymbol{\theta}}^{\star}} \quad (19)$$

Notice that $\boldsymbol{\theta}_k = \boldsymbol{\theta}_0 - \eta\sum_{i=0}^{k}\boldsymbol{g}(\boldsymbol{\theta}_i)$ and thus

$$-\mathbb{E}\left[\boldsymbol{\theta}_k\right]^{\top}\boldsymbol{F}(\hat{\boldsymbol{\theta}}^{\star})\hat{\boldsymbol{\theta}}^{\star} = -\boldsymbol{\theta}_0^{\top}\boldsymbol{F}(\hat{\boldsymbol{\theta}}^{\star})\hat{\boldsymbol{\theta}}^{\star} + \eta\mathbb{E}\left[\left[\sum_{i=0}^{k}\boldsymbol{g}(\boldsymbol{\theta}_i)\right]^{\top}\boldsymbol{F}(\hat{\boldsymbol{\theta}}^{\star})\hat{\boldsymbol{\theta}}^{\star}\right] \quad (20)$$

Then combining Eq. 19 and Eq. 20, we have

$$\frac{1}{\text{Tr}\left(\boldsymbol{F}(\hat{\boldsymbol{\theta}}^{\star})\right)} \leq \frac{M}{\mathbb{E}\left[\boldsymbol{\theta}_k \boldsymbol{F}(\hat{\boldsymbol{\theta}}^{\star})\boldsymbol{\theta}_k\right] + 2\eta\mathbb{E}\left[\sum_{i=0}^{k}\boldsymbol{g}(\boldsymbol{\theta}_i)\right]^{\top}\boldsymbol{F}(\hat{\boldsymbol{\theta}}^{\star})\hat{\boldsymbol{\theta}}^{\star} + (\boldsymbol{\theta}^{\star} - 2\boldsymbol{\theta}_0)^{\top}\boldsymbol{F}(\hat{\boldsymbol{\theta}}^{\star})\hat{\boldsymbol{\theta}}^{\star}}$$

By applying the equation above to Eq. 17, we further have

$$\text{Tr}(\boldsymbol{F}^{-1}(\hat{\boldsymbol{\theta}}^{\star})) \leq \frac{d^2 M}{\mathbb{E}\left[\boldsymbol{\theta}_k \boldsymbol{F}(\hat{\boldsymbol{\theta}}^{\star})\boldsymbol{\theta}_k\right] + 2\eta\mathbb{E}\left[\sum_{i=0}^{k}\boldsymbol{g}(\boldsymbol{\theta}_i)\right]^{\top}\boldsymbol{F}(\hat{\boldsymbol{\theta}}^{\star})\hat{\boldsymbol{\theta}}^{\star} + (\boldsymbol{\theta}^{\star} - 2\boldsymbol{\theta}_0)^{\top}\boldsymbol{F}(\hat{\boldsymbol{\theta}}^{\star})\hat{\boldsymbol{\theta}}^{\star}}$$
$$+ \frac{d\lambda_1(\boldsymbol{F}(\hat{\boldsymbol{\theta}}^{\star})) - d}{\lambda_d(\boldsymbol{F}(\hat{\boldsymbol{\theta}}^{\star}))}$$

The conclusion can be immediately obtained by applying Theorem 3. $\qquad\square$

## D    PROOF OF THEOREM 2

**Theorem** 2 *Assume A.1-A.4. Let* $\mu_k = \sum_{i=0}^{k} \mathbb{E}\left[\|\nabla\ell(\boldsymbol{\theta}_i;\mathcal{D}_i)\|_1\right]$ *denote the sum of the expected absolute value of gradient across mini-batch samples, denoted by* $\mathcal{D}_i$, *where the gradient norm is*

$$\|\nabla\ell(\boldsymbol{\theta}_i;\mathcal{D}_i)\|_1 = \sum_{j=1}^{d}\left|\nabla_{\theta_i^{(j)}}\ell(\boldsymbol{\theta}_i;\mathcal{D}_i)\right| \quad and \quad \nabla_{\theta_i^{(j)}}\ell(\boldsymbol{\theta}_i;\mathcal{D}_i) = \frac{1}{|\mathcal{D}_i|}\sum_{n=1}^{|\mathcal{D}_i|}\nabla_{\theta_i^{(j)}}\ell(\boldsymbol{\theta}_i;\boldsymbol{x}_n,\boldsymbol{y}_n).$$

*Under some regularity conditions such that* $\mathbb{E}\left[(\boldsymbol{\theta}_k - \hat{\boldsymbol{\theta}}^\star)^3\right] = o(\frac{1}{k})$, *it holds*

$$\mathcal{L}(\hat{\boldsymbol{\theta}}^\star) \geq \mathbb{E}[\mathcal{L}(\boldsymbol{\theta}_k)] - \frac{1}{2}\mathbb{E}\left[\boldsymbol{\theta}_k^\top \boldsymbol{F}(\hat{\boldsymbol{\theta}}^\star)\boldsymbol{\theta}_k\right] - \eta\mu_k\|\boldsymbol{H}(\hat{\boldsymbol{\theta}}^\star)\hat{\boldsymbol{\theta}}^\star\|_\infty - \frac{1}{2}(\hat{\boldsymbol{\theta}}^\star - 2\boldsymbol{\theta}_0)^\top \boldsymbol{H}(\hat{\boldsymbol{\theta}}^\star)\hat{\boldsymbol{\theta}}^\star + o\left(\frac{1}{k}\right).$$

*Proof.* Notice that the optimality gap is $G(\boldsymbol{\theta}_k) := \mathbb{E}[\mathcal{L}(\boldsymbol{\theta}_k)] - \mathcal{L}(\hat{\boldsymbol{\theta}}^\star)$. By applying Taylor approximation up to second order in Eq. 6 and taking expectation over the filtration , we have

$$G(\boldsymbol{\theta}_k) = \frac{1}{2}\mathbb{E}\left[\boldsymbol{\theta}_k^\top \boldsymbol{H}(\hat{\boldsymbol{\theta}}^\star)\boldsymbol{\theta}_k - 2\boldsymbol{\theta}_k^\top \boldsymbol{H}(\hat{\boldsymbol{\theta}}^\star)\hat{\boldsymbol{\theta}}^\star + \boldsymbol{\theta}^{\star\top}\boldsymbol{H}(\hat{\boldsymbol{\theta}}^\star)\hat{\boldsymbol{\theta}}^\star\right] + \mathbb{E}\left[\mathcal{O}\left((\boldsymbol{\theta}_k - \hat{\boldsymbol{\theta}}^\star)^3\right)\right]$$

$$= \frac{1}{2}\mathbb{E}\left[\boldsymbol{\theta}_k^\top \boldsymbol{H}(\hat{\boldsymbol{\theta}}^\star)\boldsymbol{\theta}_k\right] + \eta\mathbb{E}\left[\sum_{i=0}^{k}\boldsymbol{g}(\boldsymbol{\theta}_i)\right]^\top \boldsymbol{H}(\hat{\boldsymbol{\theta}}^\star)\hat{\boldsymbol{\theta}}^\star + \frac{1}{2}(\boldsymbol{\theta}^\star - 2\boldsymbol{\theta}_0)^\top \boldsymbol{H}(\hat{\boldsymbol{\theta}}^\star)\hat{\boldsymbol{\theta}}^\star + o\left(\frac{1}{k}\right).$$

Then by replacing Hessian with Fisher information matrix in the first term, the optimality gap is bounded as follows

$$G(\boldsymbol{\theta}_k) \leq \frac{E\left[\boldsymbol{\theta}_k^\top \boldsymbol{F}(\hat{\boldsymbol{\theta}}^\star)\boldsymbol{\theta}_k\right]}{2} + \eta\mathbb{E}\left[\sum_{i=0}^{k}\boldsymbol{g}(\boldsymbol{\theta}_i)\right]^\top \boldsymbol{H}(\hat{\boldsymbol{\theta}}^\star)\hat{\boldsymbol{\theta}}^\star + \frac{(\boldsymbol{\theta}^\star - 2\boldsymbol{\theta}_0)^\top \boldsymbol{H}(\hat{\boldsymbol{\theta}}^\star)\hat{\boldsymbol{\theta}}^\star}{2} + o\left(\frac{1}{k}\right) \tag{21}$$

By applying Hölder's inequality and Minkowski's Inequality, the second term in Eq. 21 is

$$\mathbb{E}\left[\sum_{i=0}^{k}\boldsymbol{g}(\boldsymbol{\theta}_i)\right]^\top \boldsymbol{H}(\hat{\boldsymbol{\theta}}^\star)\hat{\boldsymbol{\theta}}^\star \leq \mathbb{E}\left[\left\|\sum_{i=0}^{k}\boldsymbol{g}(\boldsymbol{\theta}_i)\right\|_1\right]\left\|\boldsymbol{H}(\hat{\boldsymbol{\theta}}^\star)\hat{\boldsymbol{\theta}}^\star\right\|_\infty \leq \sum_{i=0}^{k}\mathbb{E}\left[\|\boldsymbol{g}(\boldsymbol{\theta}_i)\|_1\right]\left\|\boldsymbol{H}(\hat{\boldsymbol{\theta}}^\star)\hat{\boldsymbol{\theta}}^\star\right\|_\infty.$$

Therefore, since $\mu_k = \sum_{i=0}^{k}\mathbb{E}\left[\|\boldsymbol{g}(\boldsymbol{\theta}_i)\|_1\right]$ we have

$$G(\boldsymbol{\theta}_k) \leq \frac{1}{2}\mathbb{E}\left[\boldsymbol{\theta}_k^\top \boldsymbol{F}(\hat{\boldsymbol{\theta}}^\star)\boldsymbol{\theta}_k\right] + \eta\mu_k\|\boldsymbol{H}(\hat{\boldsymbol{\theta}}^\star)\hat{\boldsymbol{\theta}}^\star\|_\infty + \frac{1}{2}(\boldsymbol{\theta}^\star - 2\boldsymbol{\theta}_0)^\top \boldsymbol{H}(\hat{\boldsymbol{\theta}}^\star)\hat{\boldsymbol{\theta}}^\star + o\left(\frac{1}{k}\right)$$

from which the conclusion immediately follows.    □

## E    PRACTICAL IMPLEMENTATION

While SiGeo necessitates a well-executed initialization in contrast to other zero-shot proxies, our experimental results indicate that the training-free SiGeo is capable of achieving comparable or superior performance to the SOTA zero-shot proxies across all computer vision tasks.

**Proxy Weights.** In all instances, we consistently set the value of $\lambda_1$ to 1. However, the selection of $\lambda_2$ and $\lambda_3$ values varies depending on the specific search space and the warm-up level. Specifically, for candidate architectures without warm-up, we adjust our proxy by setting $\lambda_2 = 1$ and $\lambda_3 = 0$ to account for significant noise in training loss values and the reduced effectiveness of the Fisher-Rao (FR) norm. Conversely, if a warm-up procedure is performed, we opt for $\lambda_2 = \lambda_3 = 1$ in RecSys search spaces, while for CV benchmarks, we use $\lambda_{50} = 50$ and $\lambda_3 = 1$. The larger value of $\lambda_2$ for CV benchmarks is due to the observation that the zico scores of CNN models are significantly larger than those of networks in the NASRec search space.

**FR Norm.** The FR norm can be efficiently computed by the average of squared product of parameters and gradients: $\boldsymbol{\theta}_k \bar{\boldsymbol{F}}\boldsymbol{\theta}_k = \frac{1}{k}\sum_{i=1}^{k}\boldsymbol{\theta}_k\nabla\ell(\boldsymbol{\theta};\mathcal{D}_i)\nabla\ell(\boldsymbol{\theta}_i;\mathcal{D}_i)^\top\boldsymbol{\theta}_k^\top = \frac{1}{k}\sum_{i=1}^{k}\left(\boldsymbol{\theta}_k^\top\nabla\ell(\boldsymbol{\theta}_i;\mathcal{D}_i)\right)^2$

# F    IMPLEMENTATION DETAILS

## F.1    EXPERIMENT (1): EMPIRICAL JUSTIFICATION FOR THEOREM 4 AND THEOREM 2

We conducted training using a two-layer MLP equipped with ReLU activation functions across the complete MNIST training dataset for three epochs. The weights were updated through the application of gradient descent as per Equation 4, employing a batch size of 128. Throughout training, these networks were optimized using the SGD optimizer with a learning rate of $0.01$.

In the preprocessing phase, all image tensors were normalized using a mean of (0.4914, 0.4822, 0.4465) and a standard deviation of (0.2023, 0.1994, 0.2010). Following these preparations, SiGeo scores were computed using four training batches after the candidate architectures were trained on 0, 48, and 192 batches, corresponding to 0%, 10% and 40% warm-up levels. Then we visualize the relationship of the training loss and test loss after three training epochs vs. (i) current training loss, (ii) FR norm and (iii) mean absolute sample gradients in Fig 5-7 in Appendix G

## F.2    EXPERIMENT (2) AND (3): COMPARING SIGEO WITH OTHER PROXIES ON NAS BENCHMARKS

To calculate the correlations, we utilize the NAS-Bench-Suite-Zero (Krishnakumar et al., 2022). This suite encompasses a comprehensive set of commonly used zero-shot proxies, complete with their official codebases released by the original authors, enabling us to accurately obtain the proxy values. Since the ZiCo proxy was not incorporated into NAS-Bench-Suite-Zero, we employed its official code to compute the correlations for ZiCo.

In the zero-shot setting, we adjust our proxy by setting $\lambda_2 = 1$ and $\lambda_3 = 0$. If a warm-up procedure is performed, we set we use $\lambda_{50} = 50$ and $\lambda_3 = 1$. The larger value of $\lambda_2$ for CV benchmarks is due to the observation that the zico scores of CNN models are significantly larger (roughly 50 times greater) than those of networks in the NASRec search space.

## F.3    EXPERIMENT (4): RECSYS BENCHMARK RESULTS

During this process, we train the supernet for a single epoch, employing the Adagrad optimizer, an initial learning rate of 0.12, and a cosine learning rate schedule on target RecSys benchmarks. We refer readers to Zhang et al. (2023) for the details about the search space, search policy, and training strategy.

**Regularized Evolution.** Following Zhang et al. (2023), we employ an efficient configuration of regularized evolution to find the optimal subnets from the supernet. Specifically, we keep 128 architectures in a population and run the regularized evolution algorithm for 240 iterations. In each iteration, we select the best design from 64 sampled architectures as the parent architecture, then create 8 new child architectures to update the population.

## F.4    EXPERIMENT (5): SIGEO V.S. OTHER PROXIES ON CIFAR-10/CIFAR-100

The experimental setup in this section mirrors that of Lin et al. (2021a), and is outlined below:

**Dataset** CIFAR-10 comprises 50,000 training images and 10,000 testing images, divided into 10 classes, each with a resolution of 32x32 pixels. CIFAR-100 follows a similar structure with the same number of training and testing images but contains 100 distinct classes. ImageNet1k, on the other hand, has over 1.2 million training images and 50,000 validation images across 1000 classes. In our experiments, we use the official training/validation split.

**Augmentation** We use the following augmentations as in (Pham et al., 2018): mix-up (Zhang et al., 2018), label-smoothing (Szegedy et al., 2016), random erasing (Zhong et al., 2020), random crop/resize/flip/lighting and AutoAugment (Cubuk et al., 2019).

**Optimizer** For all experiments in Experiment (3), we use SGD optimizer with momentum 0.9; weight decay 5e-4 for CIFAR10/100; initial learning rate 0.1 with batch size 256; cosine learning rate decay (Loshchilov & Hutter, 2017). We train models up to 1440 epochs in CIFAR-10/100.

Following previous works (Aguilar et al., 2020; Li et al., 2020), we use EfficientNet-B3 as teacher networks.

In each run, the starting structure is a randomly chosen small network that adheres to the imposed inference budget. The search for kernel sizes is conducted within the set $\{3, 5, 7\}$. Consistent with traditional design practices, there are three stages for CIFAR-10/CIFAR-100/ The evolutionary process involves a population size of 512 and a total of 480,000 evolutionary iterations. For CIFAR-10/CIFAR-100, the resolution is 32×32.

**Model Selection** During weight-sharing neural architecture evaluation, we train the supernet on the training set and select the top 15 subnets on the validation set. Then we train the top-15 models from scratch and select the best subnet as the final architecture.

## G    ADDITIONAL EMPIRICAL RESULTS FOR THEOREM 4 AND THEOREM 2

Here we conduct additional assessments in the other two distinct settings: one without any warm-up (Fig. 5) and another with various warm-up levels (Fig. 7). In the absence of warm-up, all four statistics are computed based on the initial four batches. In the case of warm-up, these statistics are calculated using four batches following the initial 196 batches.

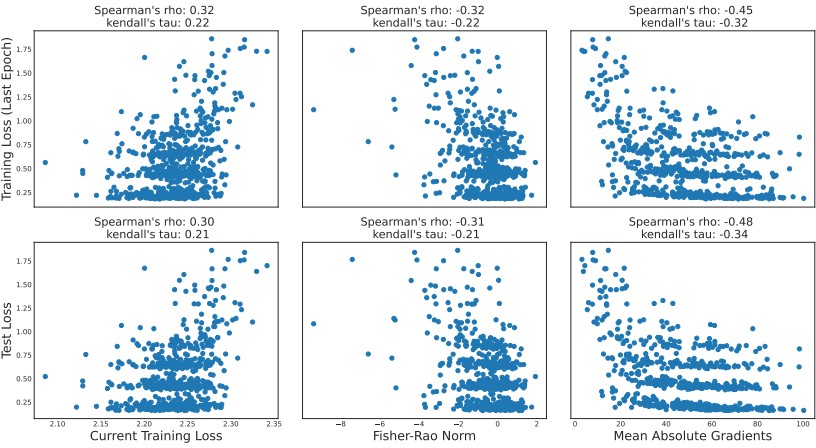

Figure 5: Training and test losses vs. different statistics in Theorem 4 and 2. Results are generated by optimizing two-layer MLP-ReLU networks with varying hidden dimensions, ranging from 2 to 48 in increments of 2. The statistics for each network are computed without warm-up.

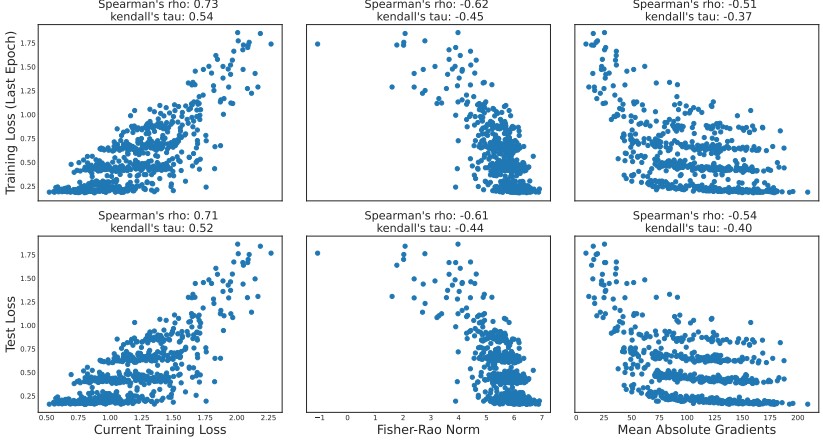

Figure 6: Training/test losses vs. statistics in Theorem 4 and 2. Results are generated by optimizing two-layer MLP-ReLU networks with varying hidden dimensions, ranging from 2 to 48 in increments of 2. The statistics for each network are computed after being warmed up with 10% training data.

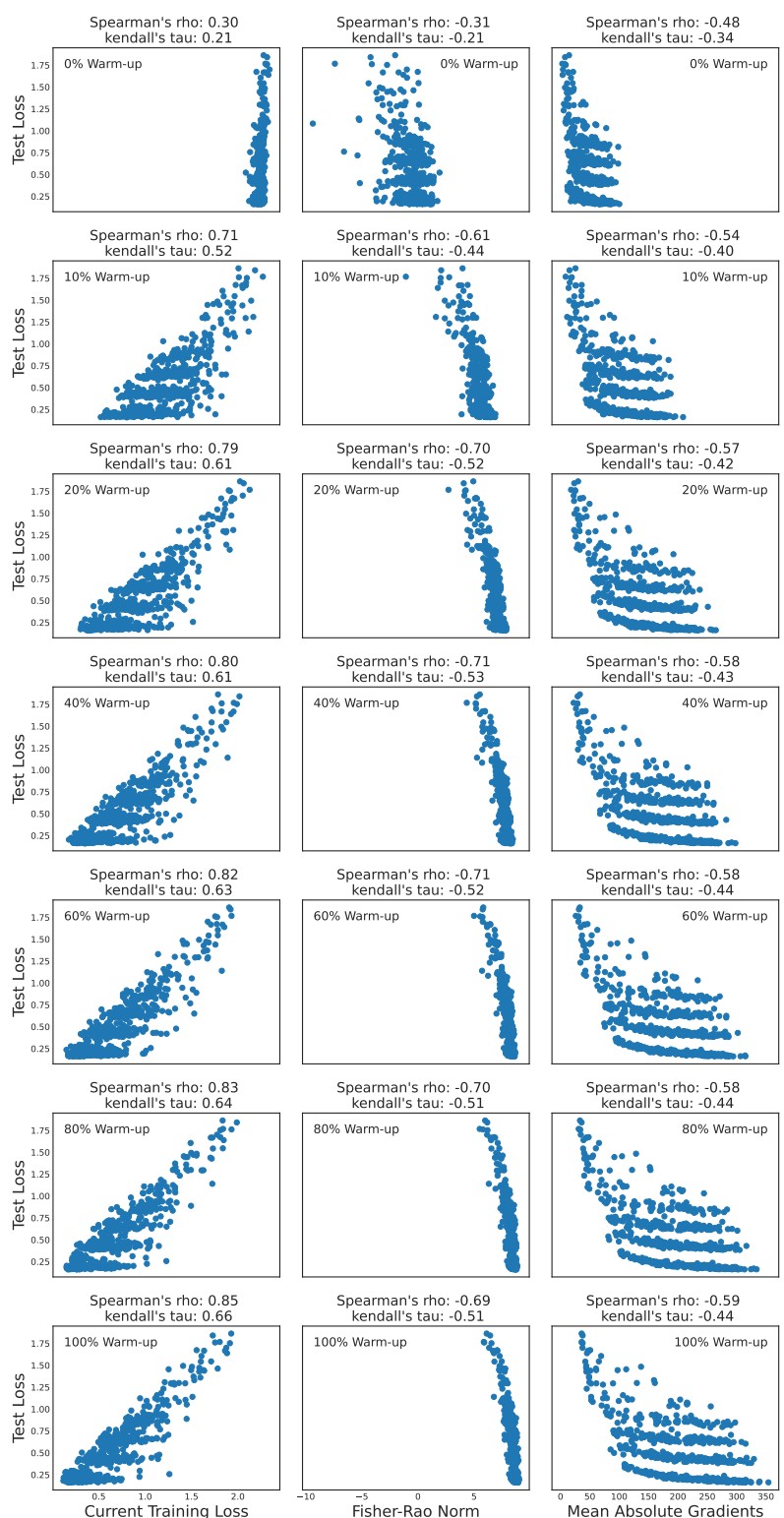

Figure 7: Test losses vs. statistics in Theorem 2. Results are generated by optimizing two-layer MLP-ReLU networks with varying hidden dimensions, ranging from 2 to 48 in increments of 2. The statistics for each network are computed after being warmed up with 0%, 10%, 40%, 60%, 80%, and 100% data. The performance improvement in the FR norm is observed to cease after 40% warm-up.

## H   SiGeo v.s. Other Proxies on CIFAR-10/CIFAR-100

Experiments on CIFAR-10/CIFAR-100 (Krizhevsky et al., 2009) are conducted to validate the effectiveness of SiGeo on zero-shot setting. This experiment serves as a compatibility test for SiGeo on zero-shot NAS. To maintain consistency with prior studies, we adopt the search space and training settings outlined in Lin et al. (2021a) to validate SiGeo across both CIFAR-10 and CIFAR-100 datasets. The selected search space, introduced in (He et al., 2016; Radosavovic et al., 2020), comprises residual blocks and bottleneck blocks that are characteristic of the ResNet architecture. We limit the network parameters to 1.0 million. Results are obtained in the *zero-shot setting*: in each trial, the subnet structure is selected from a randomly initialized supernet (without warm-up) under the inference budget; see details in Appendix F.4. Then we use a batch size of 64 and employ four batches to compute both SiGeo and ZiCo. In Table 4, we contrast various NAS-designed models for CIFAR-10/CIFAR-100, and notably, SiGeo outperforms all the baseline methods.

Table 4: Top-1 accuracies on CIFAR-10/CIFAR-100 for zero-shot proxies with a 1M parameter budge. The accuracies for both SiGeo and ZiCo are computed by averaging results from three separate runs. Other scores are adapted from Lin et al. (2021a).

| Proxy | Random | FLOPs | grad-norm | synflow | NASWOT | Zen | Zico | SiGeo |
|---|---|---|---|---|---|---|---|---|
| CIFAR-10 | 93.50% | 93.10% | 92.80% | 96.10% | 96.00% | 96.2% | 96.96% | 97.01% |
| CIFAR-100 | 71.10% | 64.70% | 65.40% | 75.90% | 77.50% | 80.10% | 81.12% | 81.19% |

## I   Additional Result for RecSys Benchmarks

We train NASRecNet from scratch on three classic RecSysm benchmarks and compare the performance of models that are crafted by NASRec on three general RecSys benchmarks. In Table 5, we report the evaluation results of our end-to-end NASRecNets and a random search baseline which randomly samples and trains models in our NASRec search space.

Table 5: Performance of SiGeo-NAS on CTR Predictions Tasks. NASRec-Small and NASRec-Full are the two search spaces, and NASRecNet is the NAS method from Zhang et al. (2023).

| Setting | Method | Criteo | | Avazu | | KDD Cup 2012 | | Search Cost |
|---|---|---|---|---|---|---|---|---|
| | | Log Loss | AUC | Log Loss | AUC | Log Loss | AUC | (GPU days) |
| Hand-crafted Arts | DLRM (Naumov et al., 2019) | 0.4436 | 0.8085 | 0.3814 | 0.7766 | 0.1523 | 0.8004 | - |
| | xDeepFM (Lian et al., 2018) | 0.4418 | 0.8052 | - | - | - | - | - |
| | AutoInt+ (Song et al., 2019) | 0.4427 | 0.8090 | 0.3813 | 0.7772 | 0.1523 | 0.8002 | - |
| | DeepFM (Guo et al., 2017) | 0.4432 | 0.8086 | 0.3816 | 0.7757 | 0.1529 | 0.7974 | - |
| NAS-crafted Arts (one-shot) (100% warm-up) | DNAS (Krishna et al., 2021) | 0.4442 | - | - | - | - | - | ∼0.28 |
| | PROFIT (Gao et al., 2021) | 0.4427 | 0.8095 | **0.3735** | **0.7883** | - | - | ∼0.5 |
| | AutoCTR (Song et al., 2020) | 0.4413 | 0.8104 | 0.3800 | 0.7791 | 0.1520 | 0.8011 | ∼0.75 |
| | Random Search @ NASRec-Small | 0.4411 | 0.8105 | 0.3748 | 0.7885 | 0.1500 | 0.8123 | 1.0 |
| | Random Search @ NASRec-Full | 0.4418 | 0.8098 | 0.3767 | 0.7853 | 0.1509 | 0.8071 | 1.0 |
| | NASRecNet @ NASRec-Small | **0.4395** | **0.8119** | 0.3741 | 0.7897 | 0.1489 | 0.8161 | ∼0.25 |
| | NASRecNet @ NASRec-Full | 0.4404 | 0.8109 | 0.3736 | 0.7905 | 0.1487 | 0.8170 | ∼0.3 |
| | SiGeo @ NASRec-Small | 0.4399 | 0.8115 | 0.3743 | 0.7894 | 0.1487 | 0.8171 | ∼0.1 |
| | SiGeo @ NASRec-Full | 0.4403 | 0.8110 | 0.3741 | 0.7898 | 0.1484 | 0.8187 | ∼0.12 |
| NAS-crafted Arts (sub-one-shot) (1% warm-up) | ZiCo @ NASRec-Small | 0.4404 | 0.8109 | 0.3754 | 0.7876 | 0.1490 | 0.8164 | ∼0.1 |
| | ZiCo @ NASRec-Full | 0.4403 | 0.8100 | 0.3762 | 0.7860 | 0.1486 | 0.8174 | ∼0.12 |
| | SiGeo @ NASRec-Small | 0.4396 | 0.8117 | **0.3741** | **0.7898** | **0.1484** | **0.8185** | ∼0.1 |
| | SiGeo @ NASRec-Full | **0.4397** | **0.8116** | 0.3754 | 0.7876 | **0.1485** | 0.8178 | ∼0.12 |
| NAS-crafted Arts (zero-shot) (0% warm-up) | ZiCo @ NASRec-Small | 0.4408 | 0.8105 | 0.3770 | 0.7849 | 0.1491 | 0.8156 | ∼0.09 |
| | ZiCo @ NASRec-Full | 0.4404 | 0.8108 | 0.3772 | 0.7845 | 0.1486 | 0.8177 | ∼0.11 |
| | SiGeo @ NASRec-Small | 0.4404 | 0.8109 | 0.3750 | 0.7882 | 0.1489 | 0.8165 | ∼0.09 |
| | SiGeo @ NASRec-Full | 0.4404 | 0.8108 | 0.3765 | 0.7856 | 0.1486 | 0.8177 | ∼0.11 |

