# OpenReview forum: "SiGeo: Sub-One-Shot NAS via Information Theory and Geometry of Loss Landscape"
_ICLR.cc/2024/Conference — Submitted to ICLR 2024_

### Official Review · Reviewer_RQP1 · 2023-10-31

**Soundness:** 3 good
**Presentation:** 2 fair
**Contribution:** 2 fair
**Rating:** 5
**Confidence:** 3

**Summary:**

This paper introduces the sub-one-shot paradigm that connects zero-shot and one-shot NAS through information theory and the geometry of loss landscapes. It proposes a proxy metric called SiGeo and a theoretical framework that connects the supernet warm-up with the efficacy of zero-cost proxy. Experimental results show that SiGeo exhibits good consistency on NAS benchmarks and performs comparably to one-shot NAS in recommendation systems.

**Strengths:**

1. SiGeo can connect the performance evaluation of zero-shot and one-shot NAS.
2. The paper provides theoretical analysis from the perspectives of convergence and generalization.
3. The authors provide theoretical empirical justification and experimental validation in the field of recommendation systems.

**Weaknesses:**

1. One concern is the novelty of the paper. SiGeo feels like a combination proxy of ZiCO, FR norm, and loss functions.

2. SiGen is Sub-One-Shot, but the authors did not use the warm-up to analyze the correlation and search accuracy in the main experiments, including NAS Benchmarks and CIFAR-10/CIFAR-100. As the authors mentioned, SiGen is equivalent to a simplified ZiCO without warm-up, so the performance improvement in Tables 2 and 3 is also marginal.

3. It would have been preferable to conduct experiments on NAS-Bench-201 benchmark and ImageNet dataset.

**Questions:**

1. It seems a bit inconsistent that SiGeo utilizes both gradient mean and variance since Theorem 1 only provides gradient variance in the bound.
2. Theorem  2 only offers a lower bound analysis. It would be better to provide an upper bound analysis of $L(\hat{\theta}^*)$.
3. In section 4.2, is there a trade-off between a longer warm-up period (e.g., 60%, 80%) and consistency of SiGeo, or does a longer warm-up period consistently improve consistency?

---

> ### Author Response · Authors · 2023-11-20
>
> Thank you for your comprehensive review and constructive feedback on our manuscript. We appreciate the recognition of SiGeo's ability to bridge zero-shot and one-shot NAS, as well as the theoretical analysis we provided.
>
> 1. **Novelty of SiGeo**: We understand your concern regarding the novelty of SiGeo. We aim to clarify that SiGeo is not merely a combination of these elements but an innovative integration that leverages each component to address specific challenges in NAS. SiGeo's novelty lies in how these elements are unified into a coherent sub-one-shot framework, offering new insights and capabilities beyond their individual applications.
>
>     - We clarify that the contribution of this work is not a new zero-shot proxy. Instead, we focus on showing the significance of the sub-one-shot and warm-up procedures, along with SiGeo's capacity to leverage information acquired as the candidate network improves with training.
>     - We clarify that the objective of our zero-shot NAS benchmark experiments was not necessarily to demonstrate superior performance to the zero-shot approaches. Rather, our aim was to validate its "backward" comparability to the zero-shot setting.
>     - We further included additional experiments; see the comment ["New Experiments"](https://openreview.net/forum?id=EcDO5EXFdH&noteId=CSjlIALRuD). To summarize, our experiments have shown an enhancement in the performance of the Fisher-Rao norm and training loss as the warm-up level increases.
>     - Our key results in Table 1 and Figure 3 demonstrate that SiGeo with a 1% warm-up of supernet can achieve significantly better performance than ZiCo and comparable performance to one-shot NAS methods with a significant reduction in computational costs.
>     - We also direct the reviewer to [General Clarification of Contribution](https://openreview.net/forum?id=EcDO5EXFdH&noteId=AwltH9CFe2).
>
> 2. **Utilization of Warm-Up in Main Experiments**: Your observation about the lack of warm-up analysis in the main experiments is well-taken.
>     - In response, we included an in-depth analysis of the correlation between warm-up periods and search accuracy in NAS benchmarks and CIFAR datasets. This will provide a clearer understanding of SiGeo's performance in various warm-up scenarios and its distinction from the ZiCo approach.
>     - We also want to emphasize the novelty of this paper lies in the coherent integration of SiGeo proxy with weight-sharing NAS.
>
> 3. **Experiments on NAS-Bench-201 and ImageNet**: We agree that including experiments on NAS-Bench-201 and the ImageNet dataset would strengthen our findings. Some of our experiments in Section 4.3 were conducted on NAS-Bench-201 benchmark and datasets.
>
> **Responses to Questions**
>
> 1. *It seems a bit inconsistent that SiGeo utilizes both gradient mean and variance since Theorem 1 only provides gradient variance in the bound.*
>
>    - **Response**: We acknowledge the need for better clarity on this aspect. The absolute value of the gradient mean was appeared in the **Theorem 2.** as part of the lower bound of the minimal achievable training loss.
>
> 2. *Theorem 2 only offers a lower bound analysis. It would be better to provide an upper-bound analysis*
>
>     - **Response**: Your suggestion for an upper bound analysis is valuable. We will explore this in our future work, thereby providing a more balanced theoretical perspective.
>
> 3. *In section 4.2, is there a trade-off between a longer warm-up period (e.g., 60%, 80%) and consistency of SiGeo, or does a longer warm-up period consistently improve consistency?*
>
>     - **Response**: As two reviewers asked the same question, we have performed a side experiment study to illustrate the consistency of SiGeo. The results have been posted in the comment above ["New Experiments"](https://openreview.net/forum?id=EcDO5EXFdH&noteId=CSjlIALRuD).

---

### Official Review · Reviewer_DixF · 2023-10-31

**Soundness:** 2 fair
**Presentation:** 3 good
**Contribution:** 2 fair
**Rating:** 5
**Confidence:** 4

**Summary:**

This paper proposes to employ sub-one-shot NAS to trade off the zero-shot  and one-shot NAS. Furthermore, this paper designs a novel proxy called SiGeo, which is based on the information	 and geometry of the loss landscape. SiGeo can achieve better performance	on CV tasks compared with zero-shot NAS and on the RecSys domain with less computational costs compared with one-shot NAS.

**Strengths:**

1. This paper provides the theoretical analysis of SiGeo by jointly considering the minimum achievable training loss and generalization error.
2. The sub-one-shot NAS framework can be better than zero-shot NAS, and can also consume	 less search cost than one-shot NAS.

**Weaknesses:**

1. The proxy is mainly borrowed from ZiCo, so the novelty is not enough.
2. The experimental results show no significant gain, especially compared with ZiCo on CV tasks.

**Questions:**

1. The theoretical verification in Figure 2 is mainly conducted on a two-layer MLP-ReLU network, so whether this theory applicable to more complex networks? For example, for the architectures in NAS-Bench-201, what about the ranking consistency when warming up with 0%, 10%, and 40% data?
2. The experiments on the CV task only consider the zero-shot settings. What about the performance when involving SiGeo proxy under the sub-one-shot NAS framework?
3. Similarly, in the RecSys experiments of Figure 3, the zero-shot NAS with SiGeo should also be compared.
4. Can the SiGeo proxy search for more complex networks on ImageNet, such as in the MobiletNet search space or the Transformer search space?
5. Since when $\lambda_2$ and $\lambda_3$ in Formula (7) are set to zero, SiGeo is simplified to ZiCo. Can other existing zero-shot proxies also improve performance by adding the two items?

---

> ### Author Response · Authors · 2023-11-20
>
> **Addressing Weaknesses:**
>
> 1. *Novelty of SiGeo*: We understand your concern regarding the novelty of SiGeo, particularly in relation to its similarities with ZiCo, Fisher-Rao (FR) norm, and loss functions. We aim to clarify that SiGeo is not merely a combination of these elements but an innovative integration that leverages each component to address specific challenges in NAS. SiGeo's novelty lies in how these elements are unified into a coherent sub-one-shot framework, offering new insights and capabilities beyond their individual applications.
>     - We clarify that the contribution of this work is **not** a new zero-shot proxy. Instead, we focus on showing the significance of the **sub-one-shot** and **warm-up** procedures, along with SiGeo's capacity to leverage information acquired as the candidate network improves with training.
>     - We clarify that the objective of our zero-shot NAS benchmark experiments was not necessarily to demonstrate superior performance to the state-of-the-art (SOTA) zero-shot approaches. Rather, our aim was to validate its "backward" comparability to the **zero-shot setting**.
>     - To better demonstrate the contribution of this work, we included additional experiments; see the comment above ["**New Experiments**"](https://openreview.net/forum?id=EcDO5EXFdH&noteId=CSjlIALRuD). It provides a more comprehensive comparison of our method against ZiCo under sub-one-shot setting in various NAS benchmarks. To summarize, our experiments have shown an enhancement in the performance of the Fisher-Rao norm and training loss as the warm-up level increases. Specifically, in the NAS-201 benchmark, the ranking correlation of the Fisher-Rao norm rose from 0.35 to 0.78 with an increase in warm-up from 0% to 20%. In contrast, the Zico proxy shows limited improvement with additional training.
>     - Our key results are presented in Table 1 and Figure 3. It demonstrates that SiGeo with a 1% warm-up of supernet can achieve comparable performance to one-shot NAS methods but with a significant reduction in computational costs. In addition, SiGeo shows significant performance improvement over the SOTA zero-shot proxy ZiCo with a negligible increase in computational time.
> 2. **Experimental Results Comparison**: We acknowledge your observation about the lack of significant gains over ZiCo in CV tasks. In response, we have conducted additional experiments to more comprehensively compare our method against ZiCo **under sub-one-shot setting**; see the results in the comment above ["**New Experiments**"](https://openreview.net/forum?id=EcDO5EXFdH&noteId=CSjlIALRuD).
>
> **Responses to Questions:**
>
> 1. *The theoretical verification in Figure 2 is mainly conducted on a two-layer MLP-ReLU network, so whether this theory applicable to more complex networks? For example, for the architectures in NAS-Bench-201, what about the ranking consistency when warming up with 0%, 10%, and 40% data?*
>
>      - **Response**: we have performed the side experiment study to illustrate the consistency of SiGeo. The results have been posted in the comment above ["**New Experiments**"](https://openreview.net/forum?id=EcDO5EXFdH&noteId=CSjlIALRuD).
>
> 2. *The experiments on the CV task only consider the zero-shot settings. What about the performance when involving SiGeo proxy under the sub-one-shot NAS framework?*
>     - **Response**: new experiments have been added to indicate the performance of SiGeo under the sub-one-shot NAS.
>
> 3. *Similarly, in the RecSys experiments of Figure 3, the zero-shot NAS with SiGeo should also be compared.*
>
>    - **Response**: The zero-shot result of SiGeo is listed in Table 5 in Appendix I. We observe that without warm-up SiGeo performs on par with ZiCo in RecSys benchmarks.
>
> 4. *Can the SiGeo proxy search for more complex networks on ImageNet, such as in the MobiletNet search space or the Transformer search space?*
>
>    - **Response**: Recognizing that the regularity assumptions of initiation, local convexity, and almost everywhere differentiability are generally easily met, we are confident in the applicability of SiGeo across various search spaces within the sub-one-shot setting. In our future work, we aim to thoroughly explore and validate this applicability. In the revised version of our manuscript, we will include preliminary findings and insights that underscore the potential and versatility of SiGeo in these broader contexts.
>
> 5. *Since when $\lambda_2$ and $\lambda_2$ in Formula (7) are set to zero, SiGeo is simplified to ZiCo. Can other existing zero-shot proxies also improve performance by adding the two items?*
>
>    - **Response**: Theoretically speaking, when the candidate architecture has been warmed up, we believe that the integration of the two terms from Formula (7) could offer improvements in other contexts as well.

---

> ### Author Response · Authors · 2023-11-21
> **Sincerely expecting further discussions with reviewer DixF**
>
> Dear Reviewer DixF,
>
> Thank you for your insightful and constructive feedback. As the discussion period is drawing to a close, we wish to make the most of this remaining time to engage in a fruitful dialogue regarding our paper. We hope you have had the opportunity to review our responses to your comments, where we endeavored to thoroughly address each of your concerns.
>
> Should you require any further information or clarifications, please do not hesitate to let us know. We are eager to provide any additional details that might assist in your evaluation.
>
> Thanks again! Authors of Paper 5974

---

### Official Review · Reviewer_Hu1G · 2023-11-04

**Soundness:** 1 poor
**Presentation:** 2 fair
**Contribution:** 1 poor
**Rating:** 3
**Confidence:** 5

**Summary:**

This paper proposes SiGeo, a new proxy to conduct neural architecture search. The main contributions include some theoretical insights, upon which the proxy is designed. Numerical experiments validate the efficacy of the proposed method.

**Strengths:**

- The paper is written well and easy to follow.
- The way of using theorem to design proxy makes sense.

**Weaknesses:**

My main concerns are

- Too strong/wrong assumptions.  The foundations of their theorems, e.g., A3-A4, are rarely held for DNNs. For examples, there exists a lot operators in the search space that results in non-differentiability. Meanwhile, the Hessian matrix being positive-definitive implies the objective function as convex at least while is non-convex for Deep learning. In addition for A2, the authors used $\ell_1$-norm to bound point in a ball, which is wrong since the domain is not a ball at all under $\ell_1$ norm.

- Wording is a bit big. The paper is titled as information theory, yet I did not find a specific area that actually leverages it significantly, except the usage of Fisher information matrix, while is commonly used as an alternative to Hessian matrix in other literatures.

- Theorem is not necessary for the proxy designs. I would expect that the theorem could indeed provide some novel insights to design some unique and innovative proxy. However, the present proxy seems quite standard for me that researchers should be able to design it out without deriving theorem under strong assumptions beforehand.

Due to the above main reasons, I could not accept the paper at the moment, though the routine of using theorem to design algorithm is what I agree on.

**Questions:**

See the weakness.

---

> ### Author Response · Authors · 2023-11-19
>
> Thank you for your detailed review and constructive feedback on our manuscript. We value your comments and would like to address weaknesses you've identified.
>
> 1. **Assumptions and Theoretical Foundations**: We acknowledge your concerns about the assumptions (A3-A4). However, we would like to offer some clarifications that might shed light on our approach:
>     - Assumption 2: The “ball” is a typo, we will change it to “set” in our revised manuscript.
>     - Assumption 3: We would like to clarify our differentiability assumption by using a rigorous definition of "differentiable almost everywhere." This means that, apart from a negligible set with zero measure, the loss function associated with candidate architectures maintains differentiability. The entire theoretical framework would still hold. In fact, the assumption is commonly implicitly assumed in the field of modern deep learning, as without it, the gradient of loss function might not exist and the foundation of gradient descent-based optimization methods would be undermined.
>     - Assumption 4: the Hessian matrix being positive-definitive locally in a compact set $\mathbb{B}$ (a small region near a local minimum) only implies the objective function is locally convex (instead of global convexity). This assumption is justified by the recent theoretical works that have proven “the loss will escape the saddle points and reside near a certain local minimum”; see the discussion in the first paragraph of Section 3.
> 2. **Title and Information Theory Application**: We appreciate your critique regarding the title's reference to information theory. The development of SiGeo was based on a significant amount of knowledge and existing work from Information theory in statistics. For example, the use of Fisher information matrix, the loss landscape, and the Fisher-Rao norm.
> 3. **Theoretical Necessity and Proxy Design**: Thank you for your thought-provoking feedback regarding the necessity of the theorem in our proxy design.
>     - Indeed, the development of a new proxy might not necessarily require a theoretical framework. However, we are of the firm belief that having such a framework greatly helps both researchers and practitioners in comprehending the scope and limits of the method's applicability. This understanding is crucial for effectively applying the proxy in various contexts and for driving further innovations in the field. We have observed that the current STOA zero-shot proxies predominately focus on convolutional neural networks (such as ZiCo [1] and ZenNAS [2]) and current theoretical frameworks often lack the generality needed to effectively encompass diverse model architectures, such as Transformers. Our approach aims to bridge this gap, offering a more flexible theoretical framework that can adapt to a broader range of architectures such as RecSys domain.
>     - The theoretical framework has been instrumental in guiding our search for a new proxy. Initially, our project nearly discarded the Fisher-Rao norm, as it seemed to offer no substantial improvement over ZiCo. However, further theoretical exploration suggested that the Fisher-Rao norm could be effective, provided the weight is close to the local minimum of the loss function. It motivates us to reassess the Fisher-Rao norm, particularly in the context of pre-warming the network. Indeed, without these theoretical insights, the potential of the Fisher-Rao norm might have been overlooked.
>
> We are committed to addressing these concerns in our revised manuscript and hope that our efforts will bring the paper closer to meeting the acceptance criteria. Thank you again for your thorough review and valuable insights.
>
> **References**
> 1. [Li, Guihong, et al. "ZiCo: Zero-shot NAS via Inverse Coefficient of Variation on Gradients." ICLR (2023).](https://openreview.net/pdf?id=rwo-ls5GqGn)
> 2. [Lin, Ming, et al. "Zen-nas: A zero-shot nas for high-performance image recognition." Proceedings of the IEEE/CVF International Conference on Computer Vision. 2021.](https://arxiv.org/pdf/2102.01063.pdf)

---

> ### Comment · Reviewer_Hu1G · 2023-11-21
>
> Thanks for the responses which resolve my concerns to some extent, while I am still concerned for the assumptions and theorems.
>
> At first, if the authors assume local convexity, it typically requires training sufficiently many epochs rather than holding in the initial stage or warm-up for a short period of time. Meanwhile, the paper is written to discuss loss function landscape, which is a concept describing the overall loss function, but the local convexity actually concentrates on a small region.
>
> Secondly, the theorems proposed in the paper are not new for me. If this is a theoretical optimization paper, theorem 1 & 2 actually serve as lemmas, but not main contributing theorems. The underlying story presented in the paper need improvements.
>
> Due to the above, I decide to keep my rating though I appreciate the authors' throughout responses.

---

> > ### Author Response · Authors · 2023-11-21
> >
> > Thank you for your further review and for acknowledging our previous responses. We appreciate your continued engagement and the opportunity to address your remaining concerns regarding the assumptions and theorems in our paper.
> >
> > 1. **Assumption of Local Convexity**: We understand your concern that local convexity typically requires extensive training rather than being a characteristic of the initial stages or a short warm-up period. We want to clarify that,
> >     - In practice, to initialize all candidate networks, we can choose large learning rates instead of “good” learning rates. For example, in experiments in Section 4.2 and 4.3 of our revised paper, we used 0.2 and 1.0 learning rates respectively to facilitate the warm-up process.
> >     - Recent theoretical results have demonstrated that gradient descent converges at a **linear rate** for **deeply overparameterized** neural networks, such as those with residual connections (ResNet) [1, 2]. Given that the supernet in weight-sharing NAS architectures typically exhibits substantial overparameterization, we anticipate—and indeed have observed in our RecSys experiments detailed in Section 4.4—that initialization can be achieved swiftly and efficiently.
> >     - We emphasize that our results are present in the context of weight-sharing NAS. In the worst scenario (full training with 100% data and sufficient epochs), our approach reduces to one-shot NAS. Therefore, as long as the warm-up is not completely ineffective, our proposed method will still show advantages.
> > 2. **Theoretical Contributions**: Regarding your views on our theorems, we understand that they may not appear novel or significant as main theorems in a theoretical optimization paper. However,
> >     - It might not be a fair comparison between theoretical optimization and NAS. The theorems in NAS serve as a tool to search for better proxies; in contrast, theorems in theoretical optimization fields provide new proof techniques, optimization insights, or better rate of convergence.
> >     - Our intention was to employ these theorems in identifying the new proxies as well as the specific conditions necessary for them to function effectively.
> >     - In Appendix C of our manuscript, we present another convergence analysis detailed in Theorems 3 and 4, which are grounded under the assumption of local strong convexity. We believe this additional set of theoretical explorations offers deeper insights and further clarifies the contribution of our approach.
> >     - We admit theorem 1 might be too simple and will change it to proposition 1. In general, we will revisit the presentation of all our theorems to better position them as integral parts of our narrative, emphasizing how they contribute to the development of our proposed proxy.
> >
> > **Reference**
> >
> > [1] [Du, S., Lee, J., Li, H., Wang, L., & Zhai, X. (2019, May). Gradient descent finds global minima of deep neural networks. In International conference on machine learning (pp. 1675-1685). PMLR.](http://proceedings.mlr.press/v97/du19c/du19c.pdf)
> >
> > [2] [Dukler, Y., Gu, Q., & Montúfar, G. (2020, November). Optimization theory for relu neural networks trained with normalization layers. In International conference on machine learning (pp. 2751-2760). PMLR.](http://proceedings.mlr.press/v119/dukler20a/dukler20a.pdf)

---

### Official Review · Reviewer_wPAs · 2023-11-05

**Soundness:** 2 fair
**Presentation:** 3 good
**Contribution:** 2 fair
**Rating:** 3
**Confidence:** 4

**Summary:**

In this paper the authors propose a zero-cost proxy for NAS that is composed by 3 terms, namely the average gradient estimate divided by the gradient standard deviation, a Fisher-Rao norm and the training loss. The first terms is the same as the ZiCo zero-cost proxy, and the 2 other terms are included as an extension that takes into consideration the training dynamics in the non-zero-cost proxy regime. The authors evaluate their method in standard image classification benchmarks, as well as some recommended systems datasets.

**Strengths:**

- The theoretical motivation for proposing SiGeo is valid.

- The paper is easy to read and well-written.

- The empirical results show comparable or better performance previous ZC proxies.

**Weaknesses:**

- Most of the paper is providing a lot of theory that is already well known in the deep learning community. Then it is using that to add a Fisher-Rao norm and training loss to the ZiCo [1] ZC proxy. This is a marginal contribution and not novel enough in my opinion for the paper to be over the acceptance threshold.

- The improvements are also marginal compared to ZiCo based on the empirical evaluations reported in the paper.

- The authors should also consider evaluating their ZC proxy on NAS-Bench-Suite-Zero [2], that contains more diverse benchmarks than the image classification ones the authors evaluated on.

- No code available.

**References**

[1] https://arxiv.org/pdf/2301.11300.pdf

[2] https://arxiv.org/pdf/2210.03230.pdf

**Questions:**

- I am puzzled how valid is the theory regarding the generalization and convergence of neural networks in the case of NAS with inheritance of the one-shot model weights. Can the authors say a few more words on this?

- What is the standard deviation of multiple runs for the reported metrics in the experiments section?

---

> ### Author Response · Authors · 2023-11-20
>
> 1. **Theoretical Background and Novelty**: We acknowledge your concerns about the theoretical aspects of our paper. We clarify our theoretical contribution:
>     - Our primary theoretical contribution is on expressing minimal achievable training loss/expected training loss in terms of gradient variance, Fisher-Rao norm, and gradient mean. It provides a foundation for our proxy and offers a justification for the necessity of network warm-up.
>     - Our theoretical results stand out in their flexibility compared to state-of-the-art (SOTA) methods. For example, the theory of [ZiCo](https://openreview.net/pdf?id=rwo-ls5GqGn) is applicable under strong assumptions of linear models or MLPs with ReLu activation functions while [ZenNAS](https://arxiv.org/pdf/2102.01063.pdf) is tailored to convolutional neural networks. In contrast, our framework requires only local convexity and differentiability almost everywhere (a.e), significantly broadening its scope.
>     -  We revised the manuscript to condense the theoretical background and focus more on the novel aspects of our work, specifically the new experiment of the Fisher-Rao norm and training loss.
> 2. **Improvement upon ZiCo Metric**: Regarding the marginal improvements compared to ZiCo, we appreciate your observation.  We aim to clarify that SiGeo is not merely a combination of these elements but an innovative integration that leverages each component to address specific challenges in NAS. In specific,
>     - We clarify that the contribution of this work is **not** a new zero-shot proxy. Instead, we focus on showing the significance of the **sub-one-shot** and **warm-up** procedures, along with SiGeo's capacity to leverage information acquired as the candidate network improves with training.
>     - We clarify that the objective of our zero-shot NAS benchmark experiments was not necessarily to demonstrate superior performance to the zero-shot approaches. Rather, our aim was to validate its "backward" comparability to the **zero-shot setting**.
>     - We further included additional experiments; see the comment ["**New Experiments**"](https://openreview.net/forum?id=EcDO5EXFdH&noteId=CSjlIALRuD). To summarize, our experiments have shown an enhancement in the performance of the Fisher-Rao norm and training loss as the warm-up level increases.
>     - Our key results in Table 1 and Figure 3 demonstrate that SiGeo with a 1% warm-up of supernet can achieve significantly better performance than ZiCo and comparable performance to one-shot NAS methods but with a significant reduction in computational costs.
>     - We also direct the reviewer to [General Clarification of Contribution](https://openreview.net/forum?id=EcDO5EXFdH&noteId=AwltH9CFe2).
>
> 3. **Evaluation on NAS-Bench-Suite-Zero**: Your suggestion to evaluate our ZC proxy on NAS-Bench-Suite-Zero is well-taken. In fact,  we indeed used NAS-Bench-Suite-Zero in our current benchmark (ref. **Section 4.3.1**).
> 4. **Code Availability**: We apologize for the absence of the code. We have attached the code as supplementary material alongside the revised manuscript.
>
> Regarding your specific questions:
>
> - **Generalization and Convergence Theory in NAS**: Our approach builds on the premise that inherited weights can provide good initial points for subnets (near some local minimum). If the inherited weight is within a **locally convex** region around a local minimum, then **Theorem 1** and **2** can be used to derive a proxy to evaluate a candidate architecture.
>     - **Theorem 1** provides an upper bound for the optimality gap, and indicates that a candidate architecture with a smaller gradient variance tends to have a smaller optimality gap.
>     - **Theorem 2** provides a lower bound for minimal achievable training loss and indicates that the minimal achievable training loss of candidate arch could be low when the expected absolute sample gradients and the FR norm are high, or the current training loss is low.
>    - Section 3.4 provides an informal analysis to clarify how the sample gradient variance is connected with the trace of Hessian and thus can serve as an indicator of a generalization error.
> - **Standard Deviation in Experiments**: Thank you for your feedback, which we value greatly. In our In the revised manuscript, we will try to incorporate the standard deviation as much as possible. However, we must also consider the significant resource demands of these experiments and the current practice. For instance, although the search is reasonably fast, validating all top-15 candidate architectures in one ResSys experiment alone requires approximately five days per A100 GPU (we have 30 experiments in total), illustrating the extensive nature of these tasks. Given these constraints and common practice, we may need to defer some of these extensive experiments to future work.
>
> We hope that these revisions and clarifications will adequately address your concerns and enhance the overall contribution of our work. Thank you!

---

> ### Author Response · Authors · 2023-11-21
> **Sincerely expecting further discussions with reviewer wPAs**
>
> Dear Reviewer wPAs,
>
> Thank you for your insightful and constructive feedback. As the discussion period is drawing to a close, we wish to make the most of this remaining time to engage in a fruitful dialogue regarding our paper. We hope you have had the opportunity to review our responses to your comments, where we endeavored to thoroughly address each of your concerns.
>
> Should you require any further information or clarifications, please do not hesitate to let us know. We are eager to provide any additional details that might assist in your evaluation.
>
> Thanks again!
> Authors of Paper 5974

---

> > ### Comment · Reviewer_wPAs · 2023-11-22
> > **Thank you for the response**
> >
> > I thank the authors for the work they put in the rebuttal. However, I still find that my main concerns are not fully addressed. For instance, NAS-Bench-Suite-Zero consists of 28 tasks in total and not only 5 used in Section 4.3.1. Moreover, reinforcing my other point, I also agree with the concerns that the other reviewers noted, in particular reviewer Hu1G. Citing them: "the theorems proposed in the paper are not new for me. If this is a theoretical optimization paper, theorem 1 & 2 actually serve as lemmas, but not main contributing theorems."
> >
> > Unfortunately, I decide to keep my score and recommend the authors to extend further the theoretical contributions of the paper and the empirical evaluation on standard benchmark suites.

---

### Official Review · Reviewer_JuJh · 2023-11-07

**Soundness:** 2 fair
**Presentation:** 2 fair
**Contribution:** 2 fair
**Rating:** 5
**Confidence:** 4

**Summary:**

The paper proposes a joint use of the zero-cost and loss metrics for NAS search, aiming to reduce the computational cost of One-Shot NAS and enhance the stability of the Zero-Cost method. The authors also improve upon the shortcomings of the ZiCo metric and validate the proposed approach through experiments on multiple datasets.

**Strengths:**

- The research motivation is clear and well-defined.
- The application of the proposed method in the field of recommender systems is commendable, addressing practical needs.

**Weaknesses:**

- The paper lacks significant innovation. The concept of Sub-One-Shot has already been mentioned in Prenas, and this paper does not bring many additional insights.
- The improvement upon the ZiCo metric seems to be inconspicuous, and the contribution appears to be limited. Moreover, the experimental results in Table.2 are very similar to those of ZiCo.
- The paper lacks a comparison with Prenas in the experimental evaluation.

**Questions:**

Please see the Weaknesses.

---

> ### Author Response · Authors · 2023-11-20
>
> Thank you for your insightful review and constructive comments on our manuscript. We appreciate your recognition of the clear research motivation and the practical application of our method in recommender systems. In response to your concerns, we would like to address the points you raised:
>
> 1. *Innovation and Relation to Prenas*: We understand your concern regarding the perceived lack of significant innovation compared to Prenas. In our revised manuscript, we will emphasize the distinct aspects of our approach.
>     - **Compared to PreNAS:** Although both methods consider combining weight-sharing NSA with zero-shot proxies. SiGeo is essentially different from PreNAS. (1) SiGeo performs the weight-sharing one-shot training to warm up the supernet first and then applies a proxy to search. In contrast, PreNAS reduces the sample space by a zero-cost selector and then performs weight-sharing one-shot training on the preferred architectures. (2) SiGeo is developed based on the idea that the warm-up of the supernet will initialize all subnets simultaneously while PreNas is developed based on the idea of using a proxy to select preferred subnets to alleviate update conflicts.
> 2. *Improvement upon ZiCo Metric*: We acknowledge your feedback regarding the incremental improvement over the ZiCo metric. While our primary contributions focus on the **sub-one-shot approach**, the SiGeo proxy, and novel experiments on RecSys models/benchmark, we recognize the importance of your concerns.
>     - It is crucial to clarify that the contribution of this work is **not** a new zero-shot proxy. Instead, we focus on showing the significance of the **sub-one-shot** and **warm-up** procedures, along with SiGeo's capacity to leverage information acquired as the candidate network improves with training.
>     - We clarify that the objective of our zero-shot NAS benchmark experiments was not necessarily to demonstrate superior performance to the state-of-the-art (SOTA) zero-shot approaches. Rather, our aim was to validate its "backward" comparability to the **zero-shot setting**.
>     - To better demonstrate the contribution of this work, we included additional experiments; see the comment above ["**New Experiments**"](https://openreview.net/forum?id=EcDO5EXFdH&noteId=CSjlIALRuD). It provides a more comprehensive comparison of our method against ZiCo under sub-one-shot setting in various NAS benchmarks.
>     - Our key results are presented in Table 1 and Figure 3. It demonstrates that SiGeo with a 1% warm-up of supernet can achieve comparable performance to one-shot NAS methods but with a significant reduction in computational costs. In addition, SiGeo shows significant performance improvement over the SOTA zero-shot proxy ZiCo with a negligible increase in computational time.
> 3. *Comparison with Prenas*: We acknowledge and appreciate the suggestion to include a comparative analysis with Prenas in our experimental evaluation. We understand the potential value this comparison could bring. However, due to the significant computational and implementation resources required, we are currently unable to incorporate this comparison into the present study. We are committed to conducting this comparison and plan to include it in our future work.

---

> ### Author Response · Authors · 2023-11-21
> **Sincerely expecting further discussions with reviewer JuJh**
>
> Dear Reviewer JuJh,
>
> Thank you for your insightful and constructive feedback. As the discussion period is drawing to a close, we wish to make the most of this remaining time to engage in a fruitful dialogue regarding our paper. We hope you have had the opportunity to review our responses to your comments, where we endeavored to thoroughly address each of your concerns.
>
> Should you require any further information or clarifications, please do not hesitate to let us know. We are eager to provide any additional details that might assist in your evaluation.
>
> Thanks again!
> Authors of Paper 5974

---

### Author Response · Authors · 2023-11-20
**New Experiments**

In response to questions from multiple reviewers, we provide a set of new experiment results to demonstrate the contribution and consistency of SiGeo.
- **Result 1: NAS benchmarks with candidate architectures being warmed up**.
we conduct additional experiments to more comprehensively compare our method against ZiCo under a sub-one-shot setting. In specific, we run an additional experiment under the same setting of **Section 4.3.1** with $\lambda_2=1$, $\lambda_3=0$ in the zero-shot setting (i.e. Warm-up Level = 0%) and $\lambda_2=50$, $\lambda_3=1$ when the network is warmed up (i.e. Warm-up Level > 0%).
|Benchmarks |               | NB101-CF10 |NB101-CF10|NB201-CF10|NB201-CF10| NB201-CF100 | NB201-CF100 | NB201-IMGNT | NB201-IMGNT | NB301-CF10 | NB301-CF10 |
|---------|---------------|------------|----------|----------|----------|-------------|-------------|-------------|-------------|------------|------------|
|Method   | Warm-up Level | Spearman   |Kendall   |Spearman  |Kendall   | Spearman    | Kendall     | Spearman    | Kendall     | Spearman   | Kendall    |
|ZiCo     | 0%            | 0.63       |0.46      |0.74      |0.54      | 0.78        | 0.58        | 0.79        | 0.60        | 0.5        | 0.35       |
|ZiCo     | 10%           | 0.63       |0.46      |0.78      |0.58      | 0.81        | 0.61        | 0.80        | 0.60        | 0.51       | 0.36       |
|ZiCo     | 20%           | 0.64       |0.46      |0.77      |0.57      | 0.81        | 0.62        | 0.79        | 0.59        | 0.51       | 0.36       |
|ZiCo     | 40%           | 0.64       |0.46      |0.78      |0.58      | 0.80       | 0.61        | 0.79        | 0.59        | 0.52       | 0.37       |
|SiGeo    | 0%            | 0.63       |0.46      |0.78      |0.58      | 0.82        | 0.62        | 0.80        | 0.61        | 0.5        | 0.35       |
|SiGeo    | 10%           | 0.68       |0.48      |0.83      |0.64      | 0.85        | 0.66        | 0.85        | 0.67        | 0.53       | 0.37       |
|SiGeo    | 20%           | 0.69       |0.51      |0.84      |0.65      | 0.87        | 0.69        | 0.86        | 0.68        | 0.55       | 0.40       |
|SiGeo    | 40%           | 0.70       |0.52      |0.83      |0.64      | 0.88        | 0.70        | 0.87        | 0.69        | 0.56       | 0.41       |

There are two key observation from the results: (1) the ranking correlation of ZiCo does not improve as the warm-up level increases; (2) the ranking correlation of SiGeo improve significantly as the warm-up level increases.

The new benchmark results are consistent with the results in **Section 4.2**, underscoring the importance of the Fisher-Rao (FR) norm and training loss terms in predicting the network performance after warm-up. During the search, we observed an increase in the correlation of FR norm and training loss with the growth of the warm-up level. Therefore, this result further validates our theory and shows its applicability to more complex networks.

- **Result 2: Consistency of SiGeo over longer warm-up period**. The following experiments are performed under the same setting as **Section 4.2**. The results demonstrate the performance of SiGeo is consistent with longer warm-up period.
| Warm-up Level | Current Training Loss | Current Training Loss | Fisher-Rao Norm | Fisher-Rao Norm | Mean Absolute Gradients | Mean Absolute Gradients |
|---------------|:----------------------|-----------------------|-----------------|-----------------|:-----------------------:|-------------------------|
| Warm-up Level | Spearman              | Kendall               | Spearman        | Kendall         |        Spearman         | Kendall                 |
| 0%            | 0.30                  | 0.21                  | -0.31           | -0.21           |          -0.48          | -0.34                   |
| 10%           | 0.71                  | 0.52                  | -0.61           | -0.44           |          -0.54          | -0.40                   |
| 20%           | 0.79                  | 0.61                  | -0.70           | -0.52           |          -0.57          | -0.42                   |
| 40%           | 0.80                  | 0.61                  | -0.71           | -0.53           |          -0.58          | -0.43                   |
| 60%           | 0.82                  | 0.63                  | -0.71           | -0.52           |          -0.58          | -0.44                   |
| 80%           | 0.83                  | 0.64                  | -0.71           | -0.51           |          -0.58          | -0.44                   |
| 100%          | 0.85                  | 0.66                  | -0.69           | -0.51           |          -0.59          | -0.44                   |

---

### Author Response · Authors · 2023-11-20
**General Clarification of Contribution**

In response to the concern raised by various reviewers, we clarify the key contributions of this work:
- **SiGeo**: Fisher-Rao norm and training loss are found both experimentally and theoretically to be increasingly effective as a NAS proxy when the candidate architectures continue to warm up. By integrating these elements with ZiCo, which excels in zero-shot settings but shows little improvement with warm-up, we have developed our proxy, SiGeo.

- **Sub-one-shot NAS**: As warming up each network separately is computationally unattainable, we employ the supernet from the weight-sharing NAS as a more efficient solution, enabling simultaneous warm-up of all candidate networks.

- **Experiments**: The experimental results show that our method achieves the comparable performance of the weight-sharing one-shot NAS with a substantial reduction in computation time.

- **Broader Impact**: The application of the proposed method in the field of recommender systems extends the impact of NAS.

---

### Meta-Review · Area_Chair_8ZVN · 2023-12-04

**Metareview:**

One-shot NAS first trains a weight-sharing supernet that optimizes the average performance in a space of architectures. Then, an optimal architecture is selected based on the weights of the supernet using validation data. The first step of training the weight-sharing supernet is time-consuming. On the other hand, zero-shot NAS aims to find the optimal architecture using some metric that can be measured on the neural network without training. While it is less costly, it is believed to be less reliable than one-shot NAS. This paper proposes “sub-one-shot” NAS, where the weight-sharing supernet is warmed up by a small subset of training data. Then, the paper proposes the SiGeo metric for the sub-one-shot NAS problem, which jointly considers the estimated achievable training loss and estimated generalization error. The proposed metric can be viewed as an interpolation between zero-shot metrics like ZiCO and one-shot NAS. Then, SiGeo is compared with several NAS baselines to evaluate its performance.

**Justification For Why Not Higher Score:**

Most reviewers leaned towards rejection. Here is a summary:
- Lack of significant innovation. Reviewers think that the idea of the paper is similar to that of PreNAS and ZiCo. I suggest better discussions on the comparison to prior work.
- Lack of theoretical contributions. Reviewers think estimating the minimal training loss cannot be justified as a significant theoretical contribution, and the derivations seem standard.
- Strong theoretical assumptions. For example, convex local loss. The authors can consider measuring the Hessian spectrum, which is common now.
- Lack of connection to information theory. For this, the authors may consider changing the title to something more specific, such as Fisher information matrix.
- Limited evaluation of the theory. Verifications of the metrics are conducted on small networks.
Reviewers suggest more and larger benchmarks. The authors can consider those. At the same time, I also recognize the difficulty of adding more experiments.
- Limited improvement in test performance compared to ZiCo. Since the goal is to improve upon zero-shot NAS, the improvement against zero-shot methods should be significant. However, whether the improvement is significant (as shown in Table 1) is subjective. I will put this as the last point.

**Justification For Why Not Lower Score:**

N/A

---

### Decision · Program_Chairs · 2024-01-16

Reject